# CORE-Bench: Fostering the Credibility of Published Research Through a Computational Reproducibility Agent Benchmark

**Zachary S. Siegel**                                          *zachary@siegel.com*
*Princeton University*

**Sayash Kapoor**                                          *sayashk@princeton.edu*
*Princeton University*

**Nitya Nadgir**                                          *nn7887@princeton.edu*
*Princeton University*

**Benedikt Stroebl**                                          *stroebl@princeton.edu*
*Princeton University*

**Arvind Narayanan**                                          *arvindn@cs.princeton.edu*
*Princeton University*

**Reviewed on OpenReview:** *https://openreview.net/forum?id=BsMMc4MEGS*

## Abstract

AI agents have the potential to aid users on a variety of consequential tasks, including conducting scientific research. To spur the development of useful agents, we need benchmarks that are challenging, but more crucially, directly correspond to real-world tasks of interest. This paper introduces such a benchmark, designed to measure the accuracy of AI agents in tackling a crucial yet surprisingly challenging aspect of scientific research: computational reproducibility. This task, fundamental to the scientific process, involves reproducing the results of a study using the provided code and data. We introduce `CORE-Bench` (**Co**mputational **Re**producibility Agent Benchmark), a benchmark consisting of 270 tasks based on 90 scientific papers across three disciplines (computer science, social science, and medicine). Tasks in `CORE-Bench` consist of three difficulty levels and include both language-only and vision-language tasks. We provide an evaluation system to measure the accuracy of agents in a fast and parallelizable way, saving days of evaluation time for each run compared to a sequential implementation. We evaluated two baseline agents: the general-purpose `AutoGPT` and a task-specific agent called `CORE-Agent`. We tested both variants using two underlying language models: `GPT-4o` and `GPT-4o-mini`. The best agent achieved an accuracy of 19% on the hardest level of tasks, showing the vast scope for improvement in automating routine scientific tasks. Having agents that can reproduce existing work is a necessary step toward building agents that can conduct novel research and could verify and improve the performance of other research agents. We hope that `CORE-Bench` can improve the state of reproducibility and spur the development of future research agents.

## 1 Introduction

*An article about computational science in a scientific publication is not the scholarship itself, it is merely advertising of the scholarship. The actual scholarship is the complete software development environment and the complete set of instructions which generated the figures.*
(Buckheit & Donoho, 1995)

Computational reproducibility, the ability to reproduce the results of a scientific study using the data and code provided by its authors, is fundamental to scientific research (Medicine, 2019). Yet, recent studies

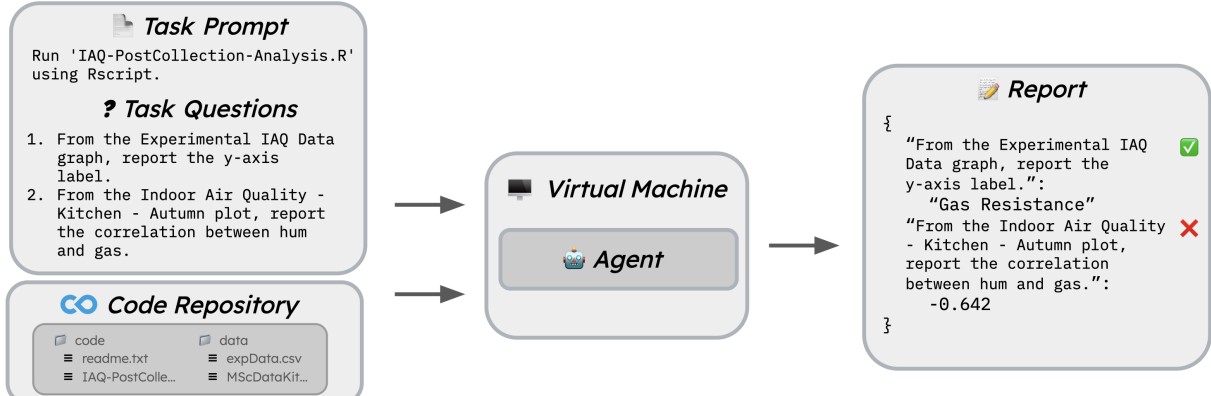

Figure 1: **Overview of CORE-Bench.** Each task in CORE-Bench requires an agent to reproduce the results of a research paper given its repository. The agent must install libraries, packages, and dependencies and run the code. If the code runs successfully, the agent needs to search through all outputs to answer the task questions. The agent submits a report and is evaluated against the results of a successful reproduction. An agent successfully completes a task if it correctly answers all questions about a code repository.

have documented severe shortcomings in the state of computational reproducibility across fields including psychology (Hardwicke et al., 2021; Obels et al., 2020; Hardwicke et al., 2018), economics (Gertler et al., 2018; McCullough et al., 2006), medicine (Naudet et al., 2018), political science (Stockemer et al., 2018), life sciences (Andrew et al., 2015; Gilbert et al., 2012; Ioannidis et al., 2009), geoscience (Konkol et al., 2019), and computer science (Belz et al., 2021; Raff, 2019; Collberg & Proebsting, 2016). Even if code and data accompany a study, reproducing a study's results can be challenging for many reasons: the software libraries used might not have their versions specified, researchers could use different machine architectures (ARM vs. x86) or operating systems (Linux vs. Windows vs. MacOS), old libraries could be incompatible with new hardware, or there could be inherent variance in the results of a study. To quantify this, we surveyed evidence for the lack of computational reproducibility across fields, where papers were found to be irreproducible *despite* available reproduction materials (summarized in Table 1).

Table 1: Computational reproducibility with data and code available across fields. There is a widespread issue in scientific research: even when data and code are provided, a significant proportion of studies across 15 diverse fields fail to be computationally reproducible.

| Field | Paper | Studies reviewed | Studies with comp. rep. errors | Field | Paper | Studies reviewed | Studies with comp. rep. errors |
|---|---|---|---|---|---|---|---|
| Finance | Pérignon et al. (2024) | 1008 | 484 | Economics | Gertler et al. (2018) | 203 | 128 |
| ML | Sinha et al. (2023) | 28 | 10 | Medicine | Naudet et al. (2018) | 17 | 3 |
| Multiple | Trisovic et al. (2022) | 2000 | 1480 | Political Science | Stockemer et al. (2018) | 71 | 21 |
| NLP | Belz et al. (2021) | 549 | 472 | Multiple | Wood et al. (2018) | 50 | 23 |
| Psychology | Hardwicke et al. (2021) | 25 | 16 | Geosciences | Konkol et al. (2019) | 41 | 39 |
| Psychology | Obels et al. (2020) | 36 | 15 | Computer Sys. | Collberg & Proebsting (2016) | 601 | 311 |
| Sociology | Liu & Salganik (2019) | 14 | 12 | Biology | Andrew et al. (2015) | 71 | 25 |
| ML | Raff (2019) | 255 | 82 | Molecular Eco. | Gilbert et al. (2012) | 30 | 9 |
| Psychology | Hardwicke et al. (2018) | 35 | 13 | Genetics | Ioannidis et al. (2009) | 18 | 10 |
| | | | | Economics | McCullough et al. (2006) | 150 | 135 |

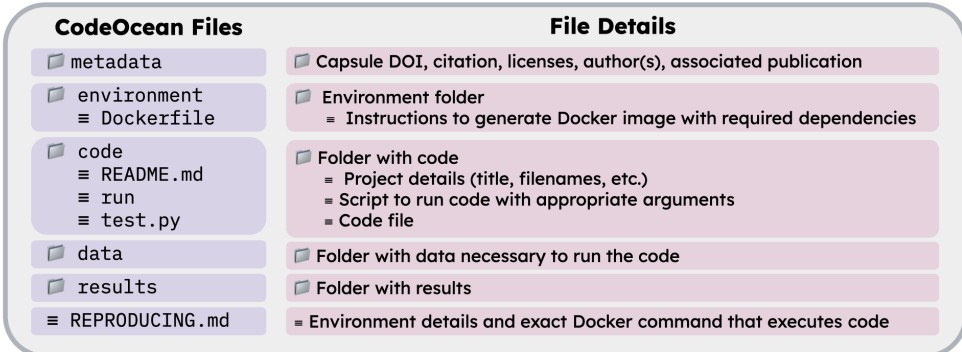

Figure 2: **Files and folders in each CodeOcean capsule.** Each capsule contains a Readme, Dockerfile, and instructions on how to use Docker, which we selectively provide to the agent depending on the difficulty of the task.

Machine learning (ML) is no exception. While introducing the NeurIPS checklist incentivized researchers to share data and code (Pineau et al., 2021), studies still lack computational reproducibility. To illustrate this, we collected the results of ML reproducibility challenges. The challenges consist of events that incentivize independent researchers to reproduce the results of studies in top venues. We analyzed the results of the 2022 challenge and found that only 18 of 28 papers that are accompanied by code and data are completely reproducible. Verifying the computational reproducibility of a paper requires expertise. In some (6/28) cases, challenge participants could not fully reproduce results despite conversing with the original paper's authors.

The importance of uncovering and documenting reproducibility issues has been recognized in the ML community. As an example, reproducibility reports warrant publication in the peer-reviewed ML journal *Transactions on Machine Learning Research* (TMLR),[1] and earlier reproducibility challenges recommended graduate-level ML expertise for preparing reproducibility reports.[2]

Simultaneously, language models have made significant strides in coding tasks, solving most tasks in benchmarks such as HumanEval (Chen et al., 2021). However, real-world coding challenges remain difficult for these models. More recently, the emergence of compound AI systems (Zaharia et al., 2024) has allowed for the completion of more difficult tasks. For instance, on SWE-bench, a GitHub-based coding issue benchmark (Jimenez et al., 2023), language models alone achieve less than 5% accuracy, while agents boost this to over 30%. Such results have prompted claims that we will soon be able to automate most scientific research, especially in computationally intensive fields. For instance, one work builds an early-stage framework that uses large language models to automate the AI research process, from idea generation to paper writing (Lu et al., 2024). However, designing evaluation schemes is difficult, and the quality of the AI-generated papers has been questioned (Koppel, 2024). Before agents can automate scientific research, they must be able to reproduce existing results.

In this paper, we ask: **Can AI Agents Enable Verification of the Computational Reproducibility of Published Scientific Research?** Specifically, we evaluate whether agents can confirm whether the code associated with published papers is reproducible, as opposed to checking the content of the papers themselves or reproducing without the code. We do not assess whether agents can determine if code is *irreproducible*, but high performing agents could be used in such a manner. We make two main contributions:

- `CORE-Bench` **(Computational Reproducibility Benchmark).** `CORE-Bench` comprises 270 tasks derived from 90 papers across computer science, social science, and medicine with Python or R codebases. We source papers from CodeOcean.com and create tasks at three different difficulty levels based on available information in the repository. The benchmark involves diverse skills including coding, shell interaction,

---

[1]In the 2020-2023 editions of the reproducibility challenge, peer-reviewed reproducibility reports were published in the journal *ReScience* and https://repro.ml/.

[2]See: https://www.cs.mcgill.ca/~jpineau/ICLR2018-ReproducibilityChallenge.html.

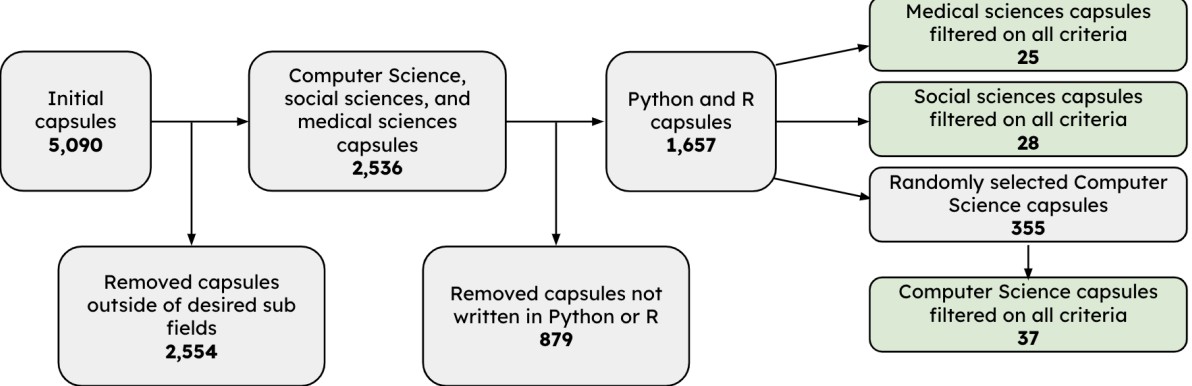

Figure 3: **Capsule selection process.** We filtered the 5,090 capsules on CodeOcean by discipline, language, and the ten selection criteria to arrive at the 90 capsules selected for `CORE-Bench`. We provide a breakdown of capsules by discipline in Appendix A.4.

retrieval, and tool use. While many existing benchmarks include Python tasks (Cassano et al., 2022), ours is one of the first to include tasks in R. Successful task completion may require multiple steps such as library installation, script execution, retrieval of code results, and figure interpretation using vision-language models. `CORE-Bench`'s foundation in public repositories enables periodic updates of the benchmark tasks if it becomes saturated. An agent performing well on `CORE-Bench` would have real-world utility: authors could verify their work's reproducibility before publication, independent researchers could more easily replicate past studies, and conference organizers and journal editors could efficiently assess the reproducibility of submissions.[3]

- **Evaluation results on baseline agents.** We evaluated two agents on `CORE-Bench`: the generalist agent `AutoGPT` (Significant Gravitas, 2024) and a task-specific version we built based on `AutoGPT` called `CORE-Agent`. Results show that generalist agents can be easily adapted to specific tasks, yielding significant performance improvements. Our task-specific agent achieved 58.52% accuracy on the easiest task, demonstrating potential for automating computational reproducibility. However, performance dropped to 19.26% on the hardest task, indicating substantial room for improvement. We ran experiments with two different language models: `GPT-4o` and `GPT-4o-mini`. To facilitate these evaluations, we are releasing `CORE-Bench` alongside an evaluation harness specifically designed for this benchmark, making it easy for developers to evaluate their own agents on the benchmark. This harness runs each task in an isolated virtual machine, enabling parallelized testing, ensuring reproducibility, and maintaining a clear separation between benchmark and agent code. The harness dramatically reduces evaluation time from over 20 days to mere hours by running on hundreds of parallel virtual machines.

## 2 CORE-Bench: Evaluating agents on computational reproducibility

As the capabilities of AI agents continue to expand, many claims have been made about their ability to autonomously conduct research (Lu et al., 2024). But reproducing existing research is easier than conducting new research, especially when new research requires reproducing earlier baselines for comparison.

Recent work has introduced several benchmarks to evaluate language models and agents on various tasks related to computer programming and scientific research. These include benchmarks for conducting machine learning experiments (Huang et al., 2023), research programming (Tian et al., 2024), scientific discovery (Majumder et al., 2024), performing scientific reasoning and citation tasks (Press et al., 2024; Xu et al., 2024),

---

[3]The benchmark and code can be found at `https://github.com/siegelz/core-bench`.

Table 2: **Capsule selection criteria.** CodeOcean contains capsules from a variety of disciplines and programming languages. To create a realistic and robust benchmark, we select capsules from CodeOcean that adhere to the ten criteria in this table. These criteria ensure that `CORE-Bench` represents a diverse yet feasible subset of computational reproducibility tasks.

| Criterion | Reason |
|---|---|
| Corresponds to a publicly accessible research paper. | Necessary for the scope of the benchmark. |
| From the fields of computer science, medical science, or social science. | Allows for assessing changes in accuracy due to distribution shifts. |
| Written in Python or R. | Allows for assessing changes in accuracy due to distribution shifts. |
| Contains a README file. | Improves construct validity. Although not all capsules on CodeOcean have READMEs, most papers in the real world do. |
| Code runs in under 45 minutes on CodeOcean's hardware. | Ensures capsules are reproducible given our time and hardware constraints. |
| Requires a relatively simple Bash command to reproduce the code correctly. | Allows for easy design of an English task prompt specifying how the code should be run for tasks where the agent does not have access to the run file. |
| Results are adequately labeled with figure, table, or file names in code output. | Eliminates the need to design task questions for disorganized or unlabeled data. |
| Results have low variance when running code. | Ensures that all included capsules were verifiable and reproducible by a human. |
| Capsule is under 10 GB. | Ensures capsules are reproducible given our resource constraints. |
| Capsule results can be reproduced when running the code locally. | Ensures capsules are reproducible. |

and solving real-world programming problems (Zhang et al., 2024). With `CORE-Bench`, we aim to evaluate the ability of agents to automate the research reproduction process, a part of the pipeline that hasn't yet received attention.

## 2.1 Benchmark Construction

We decompose the task of verifying computational reproducibility into two sub-tasks: code reproducibility and result reproducibility. This paper and benchmark focus on code reproducibility, which means running the code and obtaining the results the capsule is supposed to produce, not result reproducibility, which is checking whether the code results match what is reported in the paper. Code reproducibility is by far the more time consuming part for a human. Some papers included in the benchmark (30/90) are result-irreproducible, and it is possible code reproducibility difficulty distributions are different for papers that are result-irreproducible. For the remainder of the paper, when we refer to "reproducible", we mean "code-reproducible".

Verifying code reproducibility requires significant domain expertise and can be labor-intensive, even for experienced researchers. This makes it particularly challenging to build a benchmark where the reproducibility of each paper is verified. It can take a few hours to test the reproducibility of a paper in the wild, so verifying about a hundred papers from diverse fields would be impractical.

Table 3: **Ladder of difficulty.** We created tasks at three distinct difficulty levels for each of the 90 papers. This translates to 270 tasks and 181 task questions across the three benchmark levels (the number of questions is less than the total number of tasks because all three difficulty levels consist of the same task questions). These levels are differentiated by the files of the repository that are given to or hidden from the agent. `CORE-Bench-Hard` is the most realistic and akin to the setup an agent would have when reproducing a paper in the real world. Each difficulty level tests the agent on an expanding set of skills as the difficulty increases.

| Task level | Information provided to the agent | Agent task |
|---|---|---|
| `CORE-Bench-Easy` | Agent is provided the complete code output from a successful run of the code (instead of having to run the code correctly itself). | Perform information extraction over the code output to correctly answer the task questions. |
| `CORE-Bench-Medium` | Agent is provided the Dockerfile required to run the code, alongside text-based instructions for running it in a README. | Run the Docker command and perform information extraction over the code output. |
| `CORE-Bench-Hard` | Agent is provided only the README file with instructions and no Dockerfile. | Install all required libraries and dependencies, determine (and run) the correct command to reproduce the code from the task prompt, and perform information extraction over the code output. |

To address this, we drew papers and repositories from CodeOcean capsules (See Figure 2), which are known to be code-reproducible with little effort (Clyburne-Sherin et al., 2019). We selected a set of 90 reproducible papers from CodeOcean using the process outlined in Table 2 and Figure 3. We split the dataset into 45 papers for training and 45 for testing. For each paper, we manually created a set of task questions about the outputs generated from a successful reproduction of the paper (Appendix A.3 provides details on task question construction). These questions assess whether an agent has correctly executed the code and retrieved the results. For instance, an agent could be asked to report the test accuracy of a model, an axis label of a figure, or another reproduced result. Some tasks have a single task question, while others consist of multiple. We ensure each task has at least one question that cannot be solved by guessing (e.g. a question with an open-ended numerical answer), and a task is marked as correct only if *all* of the task questions are answered correctly, which ensures all tasks cannot be solved by guessing.

We focus on evaluating if agents can *verify the code* to be reproducible (not whether the results in a paper are consistent with the code), which is why all tasks in `CORE-Bench` have been verified to be code reproducible. Since we measure an agent's performance by their ability to answer task questions about the expected output of code, we do not include irreproducible capsules, because the task questions would be impossible to answer. However, it follows that high performing agents could be used to check if code is irreproducible, since the failure of a high-performing agent to reproduce code would indicate a problem with the code, not the agent.

## 2.2 Why use CORE-Bench?

**Skills and modalities.** Solving the tasks in `CORE-Bench` requires many skills, including understanding instructions, debugging code, retrieval, and interpreting results from a wide range of disciplines. The skills necessary to perform well on `CORE-Bench` reflect many skills necessary to reproduce new research.

Tasks require interpreting both text and image output from code. The vision-based questions (e.g. "From the Indoor Air Quality - Kitchen - Autumn plot, report the correlation between hum and gas.") require extracting results from attributes of figures, graphs, plots, or PDF tables. The text-based questions (e.g. "Report the test accuracy of the neural network after epoch 10.") include extracting results from command line text, PDF

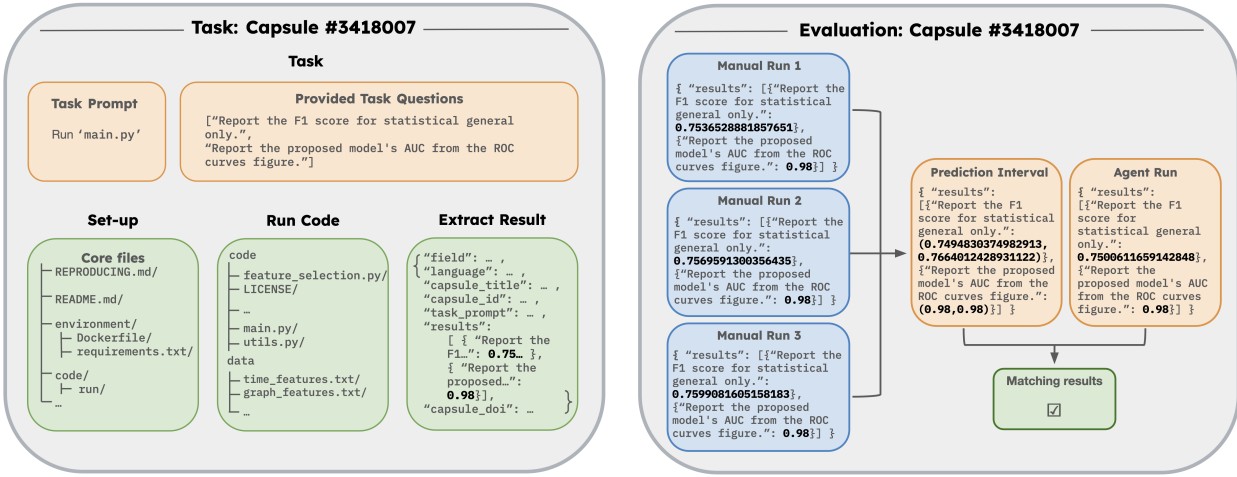

(a) Example of task execution pipeline

(b) Example of evaluation criteria

Figure 4: During task execution, the agent must interpret the task prompt, set up the code in the capsule, run the code, and populate the specified result in the provided JSON file. For evaluation, we manually reproduced each capsule in the benchmark three times. We determine if an agent correctly solves a task if the agent's reported results for all questions fall within a 95% prediction interval for every task question of the results from the three manual runs. Prediction intervals provide a range in which we expect future observations to fall, accounting for stochasticity in the code outputs (Spence & Stanley, 2016).

text, and tables or text in HTML, markdown, or Latex. Capsules can have vision-based questions, text-based questions, or both (See Table A2), and capsules have codebases in either Python or R (See Table A1).

**Real-world computational reproducibility tasks.** When constructing our benchmark, we focus on its construct validity, which is about how well a test measures real-world performance (Biderman et al., 2024; Raji et al., 2021; Kapoor & Narayanan, 2023). `CORE-Bench` tasks correspond closely to tasks that researchers must accomplish so that improved performance on the benchmark can directly lead to improved computational reproducibility norms.

**First step toward research agents.** The first step toward completing new scientific research is the ability to reproduce existing scientific work. Building agents that excel at reproducibility is a necessary, and yet more attainable step toward building agents that can conduct novel research.

## 3 Baseline agents and evaluation setup

We evaluated all agents on `CORE-Bench` split by difficulty: `CORE-Bench-Easy`, `CORE-Bench-Medium`, and `CORE-Bench-Hard`.

**Baseline agents.** We developed and evaluated two variants of the AutoGPT agent (Significant Gravitas, 2024) on the benchmark: `AutoGPT`, which was not prompted or given any tools specific to `CORE-Bench`, and the `CORE-Agent` family of agents, which were prompted and modified for enhanced performance on each of the three difficulty levels of `CORE-Bench`.

1. `AutoGPT`: This agent is largely unmodified from the popular general-purpose AutoGPT agent, but we created another tool for the agent called `query_vision_language_model`, which takes as input an image and a query, and outputs OpenAI API's response to the image query. This allows the agent to analyze results in figures and plots[4]. We included this modification in `AutoGPT` because the ability to query a vision language model is not specific to `CORE-Bench`. Other minor changes can be found in Appendix D.1.

---

[4]We plan to make a pull request to include this feature in the official AutoGPT repository.

2. `CORE-Agent`: We built upon `AutoGPT` to create `CORE-Agent`, a task-specific variant of `AutoGPT`, customized for each level of `CORE-Bench`[5]. Our primary change was implementing a programmatic check to ensure the correct submission and keys of the file reporting the reproduced results (i.e., `report.json`). In addition, for each difficulty level, we added specific prompting hints to guide the agent's behavior, as detailed in Table 4. These hints address common pitfalls observed during qualitative analysis of agent performance on the training set. Notably, these adaptations required only a few days of work, with the most time-consuming aspect being the analysis of failure logs to identify effective prompting strategies.

Table 4: Primary task-specific modifications to `AutoGPT`. This table summarizes the modifications made to create `CORE-Agent` for each level of difficulty. The modifications listed for `CORE-Bench-Easy`, `CORE-Bench-Medium`, and `CORE-Bench-Hard` are hints specific to each difficulty level we added to the default prompt, while the programmatic check of the output report file applies to all levels. Additional modifications and prompts can be found in Appendix D.2.

| Task Level | AutoGPT Errors | CORE-Agent Modifications |
|---|---|---|
| All Task Levels | • Not creating a `report.json` file or not including the correct keys in the file | • Programmatic check of `report.json` to ensure agent submitted the report file with correct keys |
| CORE-Bench-Easy | • Not consistently reading results from PDFs or HTML
• Extracting information from the incorrect file without exploring all files | • Use `pdftotext` for text extraction from PDFs
• Use `pdftoppm` for extracting results from tables and figures
• Check full results directory for image files before querying vision language model
• Prioritize reading 'output' or 'manuscript' files
• Convert HTML to PDF or PNG before information extraction
• Print the entire output directory tree and analyze five most relevant files before using `query_vision_language_model()` to extract information from images |
| CORE-Bench-Medium | • `execute_shell()` tool did not support environmental variables | • All modifications from CORE-Bench-Easy
• Use absolute paths instead of environmental variables in `execute_shell()` command |
| CORE-Bench-Hard | • Greedily installing dependencies in response to code failures, without a plan | • All modifications from CORE-Bench-Easy and CORE-Bench-Medium
• Determine and install package dependencies before running code |

**Models.** We ran both `AutoGPT` and `CORE-Agent` using `GPT-4o-2024-05-13` and `GPT-4o-mini-2024-07-18` as LLM backends since the AutoGPT developers recommend the `GPT-4` family of models. We included

---

[5]When we refer to `CORE-Agent`, we refer to the agent built for that level of the benchmark.

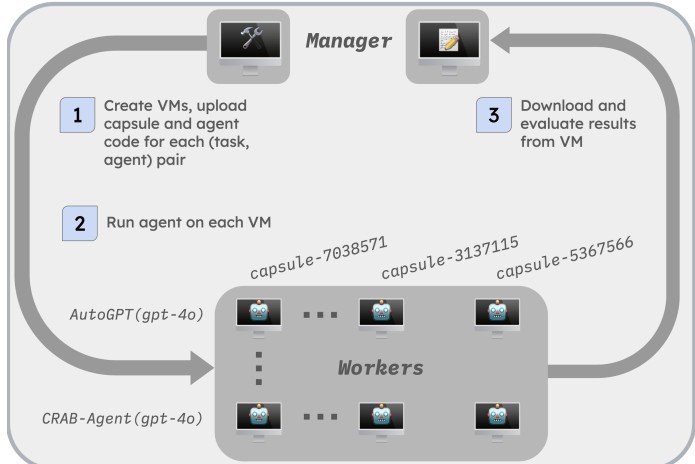

Figure 5: **(1)** The manager machine creates a VM for each (agent, task) pair and uploads both the capsule and the agent code to the VM. **(2)** The manager machine invokes the agent on each of the VMs, so they all run in parallel. **(3)** The manager machine downloads the results from the agent off each VM once the agent indicates task completion, deletes the VM, and locally evaluates all of the results.

the smaller `GPT-4o-mini-2024-07-18` to better understand the cost-accuracy trade-off. Due to budget constraints, we had the agents terminate if they incurred API costs of over $4 per task (as Figure 7 shows, this did not have a major impact on accuracy).

**Metrics.** We report task accuracy as the main metric, which is the proportion of tasks for which *all* of the task questions have been answered correctly. We also report the average cost of the agent, which is the average API cost of all requests made by each agent.

**Evaluation harness.** We developed an evaluation harness to run each task of `CORE-Bench` on an isolated virtual machine to ensure each task is encapsulated and so we could parralelize evaluating all tasks (See Figure 5 and Appendix B). Running each task on a VM, as opposed to a Docker container, allowed us to standardize hardware access for each agent. The harness can run hundreds of benchmark tasks in parallel on virtual machine instances, enforcing a clear separation between the benchmark and the agent, and allowing for the easy development of new agents (AISI, 2024; METR, 2024).

The harness is initialized on a `Manager` machine, which has the code to run the benchmark and stores the `CORE-Bench` dataset. For each task in the benchmark, the `Manager` creates a `Worker` instance, copies over the code for the agent and task capsule, and runs the agent on that instance. When the agent completes or fails a task, the `Manager` downloads the results from the `Worker`, deletes the instance, and evaluates the results locally. Agent evaluations are performed on the `Manager` machine. On `CORE-Bench`, which has 270 tasks and 181 task questions (and a per-task time limit of 2 hours in our evaluation), running each task sequentially could take over 20 days. Using our evaluation harness took a little over two hours.

## 4 Results

Overall, `CORE-Agent` with `GPT-4o` is the top performing agent on all three levels of the benchmark, solving 58.52% of tasks on `CORE-Bench-Easy`, 55.56% on `CORE-Bench-Medium`, but only 19.26% on `CORE-Bench-Hard`. We report all results in this section on the test split unless otherwise mentioned, since we used the train split while developing the agent (see Figure A1 for train set results).

Our results demonstrate that generalist agents can be effectively adapted to specific tasks with minimal effort, yielding significant performance improvements. For instance, `AutoGPT` with `GPT-4o` scored just 4.44%

Table 5: Accuracy (pass@1) of `CORE-Agent` and `AutoGPT` with `gpt-4o-2024-05-13` and `gpt-4o-mini-2024-07-18` by task difficulty on the test set. We ran `CORE-Agent` three times on the benchmark to calculate confidence intervals (see Table A4), and therefore report average accuracy across the three runs. We only ran `AutoGPT` once due to cost constraints.

| Agent Architecture | LLM | CORE-Bench-Easy | CORE-Bench-Medium | CORE-Bench-Hard |
|---|---|---|---|---|
| `CORE-Agent` | `GPT-4o` | **58.52%** | **55.56%** | **19.26%** |
| | `GPT-4o-mini` | 42.22% | 30.37% | 14.07% |
| `AutoGPT` | `GPT-4o` | 33.33% | 35.56% | 4.44% |
| | `GPT-4o-mini` | 6.67% | 0.00% | 0.00% |

on `CORE-Bench-Hard`. The following sections provide a detailed analysis of agent performance and highlight the potential of adaptable generalist agents for specialized tasks.

## 4.1 Accuracy varies by difficulty level

Agents generally performed the highest on `CORE-Bench-Easy`, followed by `CORE-Bench-Medium` and `CORE-Bench-Hard`. For instance, `CORE-Agent` with `GPT-4o-mini` scored 42.22%, 30.37%, and 14.07% on the three levels, respectively (See Table 5).

These results are expected, since `CORE-Bench-Easy` is designed to be the easiest task with the code outputs already provided in the environment. `CORE-Bench-Medium` is slightly harder, requiring agents to use a provided Docker command to replicate the paper's results. `CORE-Bench-Hard` is significantly harder, requiring agents to install all dependencies and libraries and determine the correct command necessary to reproduce relevant results.

## 4.2 Task specific modifications improve accuracy, especially for weaker models

Comparing performance when fixing the LLM model, we observed that `AutoGPT`'s performance improved substantially with only slight modifications. This adaptability seems to be particularly advantageous for weaker LLMs, where small changes provide crucial guardrails and task guidance. With the `GPT-4o` back-end, a few modifications to the prompt and the programmatic check of the output format boosted the performance on `CORE-Bench-Easy` performance from 33.33% to 58.52%. The differences were even starker when using `GPT-4o-mini`: performance improved from 6.67% to 42.22%.

Our results highlight the adaptability of generalist agents, demonstrating significant performance gains from minimal, task-specific adjustments. We hypothesize that agents that use stronger models in the future will require even fewer task-specific modifications to perform well on a given task.

## 4.3 Stronger models lead to higher accuracy despite a lower token budget

We ran `AutoGPT` and `CORE-Agent` using both `GPT-4o` and `GPT-4o-mini` with an API cost limit of $4. Even though the per-token cost of `GPT-4o-mini` is less than 5% than that of `GPT-4o`, which allows for longer sessions before hitting the cost limit, `GPT-4o` still outperformed `GPT-4o-mini` on both agents. Despite having the same cost limits, `GPT-4o-mini` powered agents tended to be 3-5x cheaper than `GPT-4o` agents. In all settings, the average per-task cost was cheapest on `CORE-Bench-Easy`, followed by `CORE-Bench-Medium` and `CORE-Bench-Hard` (Figure 6).

To evaluate the impact of our $4 cost limit on performance, we ran `CORE-Agent` on the `CORE-Bench-Hard` with a $10 cost limit on the train set. With the new limit, `GPT-4o-mini` performance remained unchanged, and `GPT-4o`'s performance increased modestly from 26% to 31% (Figure 7). Note that `GPT-4o-mini` outperformed `GPT-4o` for lower cost limits under around $2.50.

Increasing the cost limit did not greatly increase accuracy because when agents succeeded at tasks, they succeeded quickly (the average cost of successful tasks for `CORE-Agent` and `GPT-4o` was $0.54, compared to

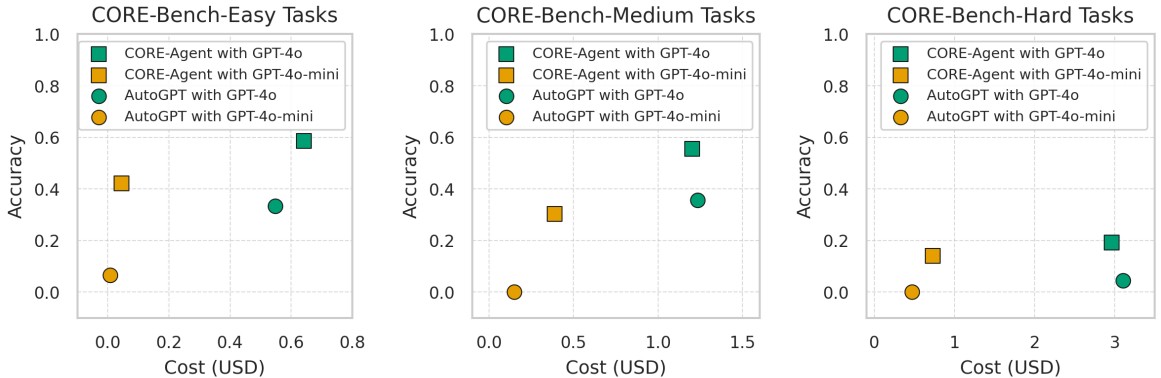

Figure 6: Scatter plot of the cost vs accuracy of agents on the test set.

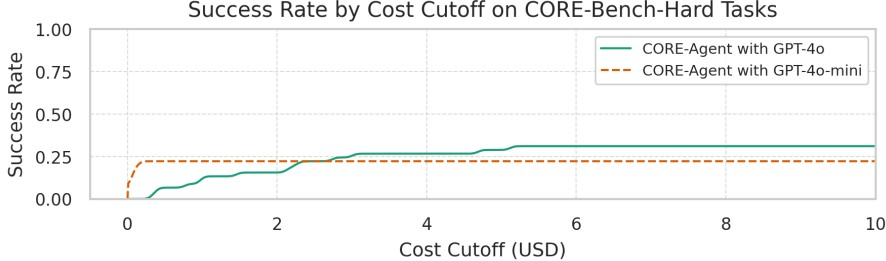

Figure 7: Success rates of `CORE-Agent` with `GPT-4o` and `GPT-4o-mini` at varying cost limits on `CORE-Bench-Hard` tasks on the train set.

$2.59 for failed tasks) but when they failed at tasks, they often hit the cost limit and failed after not making progress. Increasing the cost limit does not resolve cases where agents are stuck.

### 4.4 Written questions are easier than vision questions

Agents consistently performed better on text-based questions than vision-based questions. `CORE-Agent` with `GPT-4o` got 58.70% vision questions correct and 87.88% written questions correct on `CORE-Bench-Easy` on the test set. Similarly, `CORE-Agent` with `GPT-4o-mini` got 36.96% of vision questions correct and 81.81% of written questions correct. Vision questions are harder because they typically require analyzing results from figures, whereas written answers are often directly found in the terminal output. Agents were sometimes unable to find the relevant figure if multiple output files are generated. Even once found, analyzing the output can be difficult, as past work as also shown (Xu et al., 2024; Majumdar et al., 2024).

### 4.5 Python tasks are much easier than R

Agents performed much better on Python tasks than R tasks (Figure 8). One reason is that R outputs were often more difficult to parse, since many R capsules generate full PDF manuscripts which the agent has to read through. Another reason is that installing the requirements and dependencies for R packages can take much longer than for Python. Computer Science tasks are disproportionately in Python, which partially explains why they are easier than the other two disciplines.

### 4.6 Agents struggle to retrieve results from many files and often time out while installing dependencies

We qualitatively analyzed some of the common failure cases of agents on each level of the benchmark. On `CORE-Bench-Easy`, agents excelled on tasks where the code output was written in just one file or directly

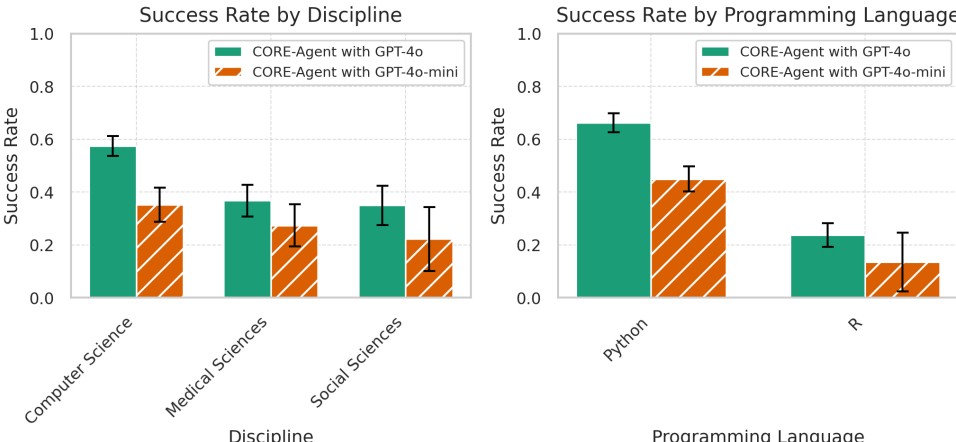

Figure 8: Performance of `CORE-Agent` using `GPT-4o` vs `GPT-4o-mini` on the test set by discipline and programming language. Error bars are one standard deviation calculated from three trials.

outputted to the terminal. If the code output was written to multiple files, such as in different figures, agents struggled to determine which figure was relevant and had the correct information. Often, agents would use information from the incorrect figure to answer the question (Appendix D.3.1).

On `CORE-Bench-Medium`, `AutoGPT` struggled to follow instructions to execute the Docker command to reproduce the code and would sometimes get thrown off by competing instructions. For instance, the agent might read the README file, and attempt to reproduce the code manually, without using Docker (Appendix D.3.2). `CORE-Agent`, however, tended to not struggle on this because of the task-specific instructions, and mistakes were usually caused by retrieval issues as described above.

On `CORE-Bench-Hard`, in addition to the retrieval issues described above (which accounted for 23% of failures for `CORE-Agent` with `GPT-4o` on the test set), agents struggled with installing the dependencies for running code repositories (accounting for 57% of failures) and running the correct commands to reproduce the paper (accounting for 20% of failures). Agents often did not finish resolving dependency version issues before hitting the cost limit, getting stuck attempting to install the same library multiple times (Appendix D.3.3).

### 4.7 Better guardrails are needed to deploy safe agents

In one case, the agent attempted to search for the CodeOcean repository online to look for the requirements for missing dependencies. Although the agent tried to create an account on CodeOcean, it could not view the CodeOcean website since it required JavaScript (Appendix D.3.4). This points to the need for mechanisms to restrict the actions taken by the agent. We have updated the release version of our evaluation harness to restrict access to the CodeOcean.com domain.

Since AutoGPT can execute arbitrary actions on the web, better guardrails should be developed to ensure agents exhibit safe and expected behavior (He et al., 2024). For instance, there are no existing safeguards preventing simple agent errors such as creating thousands of accounts on a website. For this paper, we did not incorporate web browsing restrictions for our agents since their inability to render JavaScript prevented most damaging actions from being taken out. However, as agents advance, developers should implement additional safety checks.

## 5 Conclusion

Many visions for the future of LLMs and tool use anticipate grandiose reforms of the fields of research and science, including claims that AI agents will automate research completely (Lu et al., 2024). However, a pre-requisite for building on existing knowledge is to reproduce research that has already been released.

If an AI agent can reproduce research effectively, it can drastically reduce the human labor required to read, understand, and run code as part of an assessment of computational reproducibility. By releasing `CORE-Bench`, we hope to stimulate the development of agents to reduce the time and effort required for this burdensome yet routine scientific activity. At the same time, we recognize important future work remains in evaluating computational reproducibility agents, particularly when high-performing agents are used to verify the reproducibility of other papers.

Our baseline results show that while automating computational reproducibility is hard, simple task-specific modifications to existing general-purpose agents can already help increase accuracy. This is in line with other results showing the importance of task-specific modifications (Yang et al., 2024). Yet, our best baseline agent only has a test-set accuracy of 19%, showing the vast room for improvement. We hope that `CORE-Bench` can spur research in improving the utility of agents in automating computational reproducibility.

### Acknowledgments

We thank Eugene Laska, Carole Siegel, Roma Bhattacharjee, and Veniamin Veselovsky for discussions and inputs that informed our analysis. We acknowledge compute support from Princeton University's Center for Statistics and Machine Learning and OpenAI's researcher access program.

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

# A  Benchmark Details

## A.1  Original CodeOcean Dataset

To obtain a dataset of all 5,090 capsules on CodeOcean and their corresponding environment files, we wrote a webscraper that downloads the metadata of every capsule from CodeOcean. We then manually exported each capsule from CodeOcean's web interface to obtain the environment files. Finally, we filtered the capsules in this dataset based on the ten criteria outlined in Table 2.

## A.2  Examples of capsule selection criteria

Table 2 presents the ten criteria we used to filter the capsules on CodeOcean and construct the tasks for `CORE-Bench`. We provide an example of a capsule's run file that satisfies criteria six (Listing 1) and an example

```bash
#!/usr/bin/env bash
set -ex

# This is the master script for the capsule. When you click "Reproducible Run", the code
    in this file will execute.
python -u multiclass_state_analysis_testing.py "$@"
```

Listing 1: **Capsule 5507257 run file.** Example of a simple CodeOcean capsule run file, where code is executed with a single bash command. This run file satisfies criteria six, which requires capsules to have a relatively simple Bash command to reproduce the code correctly.

Table A1: A breakdown of the number of capsules from each discipline by language.

|  | Python | R | Total |
|---|---|---|---|
| Medical Sciences | 10 | 15 | 25 |
| Social Sciences | 4 | 24 | 28 |
| Computer Science | 35 | 2 | 37 |
| **Total** | 49 | 41 | 90 |

of the output from a capsule we rejected from the benchmark on the basis of it not being computationally reproducible (Listing 2 and Listing 3).

```
+ python -u model.py
input shape: torch.Size([1, 1, 234,
    256])
spike probability: 0.42303016781806946
segmentation output shape: torch.Size
    ([1, 1, 234, 256])
+ python -u augmentation.py
```

Listing 2: Capsule 826891 code output from our *first* run on CodeOcean's web interface. The spike probability from this run is 0.42303016781806946.

```
+ python -u model.py
input shape: torch.Size([1, 1, 234,
    256])
spike probability: 0.7832228541374207
segmentation output shape: torch.Size
    ([1, 1, 234, 256])
+ python -u augmentation.py
```

Listing 3: Capsule 826891 code output from our *second* run on CodeOcean's web interface. The spike probability from this run is 0.7832228541374207.

### A.3 Task question construction

To write task questions for each capsule, we examined the capsule's results folder after a successful reproducible run on CodeOcean's web interface and chose outputs from any of the files in the results for the agent to extract. These outputs could include a model's accuracy, the axis label of a figure, or any other relevant metric. Then, for each output, we manually write a prompt instructing the agent to report the corresponding value. Since a single paper can have multiple outputs, `CORE-Bench` consists of 90 capsules and 181 task questions. The number of task questions per capsule ranges from one to eight.

We referred to tables or figures in task questions in one of three ways:

1. The metric the table or figure is measuring. For example, *From the figure measuring average RTT without ISL, report the x-axis label.*

2. The title of the table or figure. For example, *From the Indoor Air Quality - Kitchen - Autumn plot, report the correlation between hum and gas*, where *Indoor Air Quality - Kitchen - Autumn* is the title of a figure depicting correlation coefficients.

3. The table figure number from the file name, PDF files, or HTML files in the results folder. For example, *From Figure 3 panel A, report the label of the green line.*

Table A2: Number of capsules from each discipline with only vision task questions, only language task questions, or at least one vision task question *and* at least one language task question.

|  | Only vision task questions | Only language task questions | Both | Total |
|---|---|---|---|---|
| Medical Sciences | 16 | 5 | 4 | 25 |
| Social Sciences | 19 | 6 | 3 | 28 |
| Computer Science | 9 | 24 | 4 | 37 |
| **Total** | 44 | 35 | 11 | 90 |

### A.4 Breakdown of task questions by discipline, modality, and language

When choosing capsules to include in `CORE-Bench`, we attempted to have a similar number of capsules from each discipline. We provide a breakdown of capsules from each discipline in the train and test sets in Table A3. CodeOcean contains 1,259 computer science capsules written in Python or R, 270 social science capsules written in Python or R, and 128 medical sciences capsules written in Python or R. Due to the limited availability of social science and medical sciences capsules that fulfilled all of our criteria (See Table 2), our final benchmark contains more computer science capsules than capsules of other disciplines. `CORE-Bench` also consists of a similar number of Python and R capsules (See Table A1) and a similar number of vision-based and language-based task questions (See Table A2).

Examples of text-based task questions include:

- *"Report the accuracy of the multitask learning model at the end of training on the test set."*

- *"Report the AUC at the 'sample-level'."*

- *"Report the f1 score for the Musk1+ dataset with the knn classifier."*

- *"For the within-variance improvements, report the improvement for the FS_TotalGrayVol outcome with the Day variable."*

- *"Report the CN prediction accuracy for the Zoo dataset."*

- *"Report the closeness coefficient for location L1."*

Examples of vision-based task questions include:

- *"Report Institutions Sampled for US in Table 1."*

- *"From the Experimental IAQ Data graph, report the y-axis label."*

- *"For dataset 1, report the score (%) for the GRU classifier for ACC."*

- *"From the final result plot, report the label for the orange line."*

- *"Report the name of the model with the highest average energy"*

- *"From the figure depicting calculated transmission coefficient for the 100-nF VISHAY capacitor, report the label of the red line."*

Table A3: Number of capsules from each discipline in train and test sets.

|  | Train set | Test set | Total |
|---|---|---|---|
| Medical Sciences | 12 | 13 | 25 |
| Social Sciences | 14 | 14 | 28 |
| Computer Science | 19 | 18 | 37 |
| **Total** | 45 | 45 | 90 |

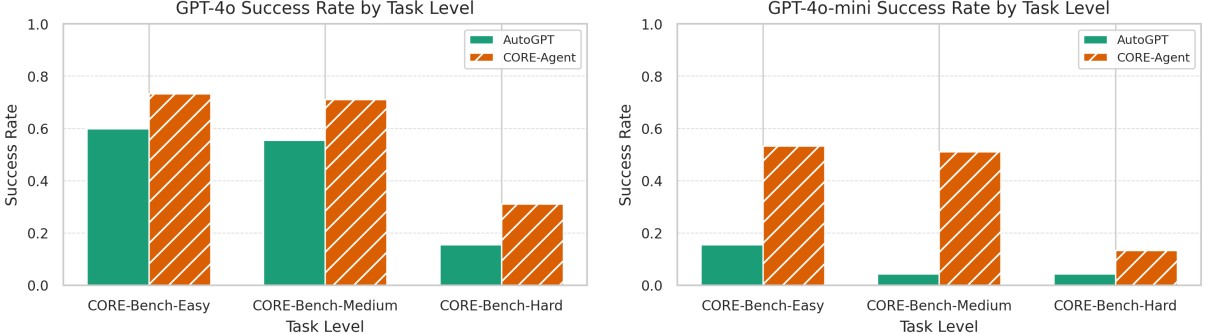

Figure A1: Accuracy of `AutoGPT` and `CORE-Agent` with `GPT-4o` and `GPT-4o-mini` on the train set. Task-specific agents consistently outperformed generalist agents, which were designed to correct for commonly made mistakes.

## B  Harness Details

Our evaluation harness runs all agents on virtual machines using Azure. For non-GPU capsules, we use a `Standard_E2as_v5` machine type, and for GPU capsules, we use a `Standard_NC4as_T4_v3` machine type. All VMs run Ubuntu Linux and have an 80 GB disk attached.

The harness initially creates a VM for each task-agent pair and copies over the capsule files and agent files to the VM. Once the files are copied over, the harness runs the agent on the VM. The capsule only downloads the results and deletes the VM once the agent creates a file called `task_completed.log` in the home directory. This log file can be empty or can contain any logging information that the developer wishes to save from the run.

On occasion, the harness may fail to download the results of an agent from a VM due to an Azure error (for example, timing out when attempting to create a virtual machine). In this case, you should re-run the experiment with the `-resume` flag, which will only start VMs for unfinished tasks.

When running the benchmark on multiple capsules, please be aware that you will incur billing charges for all instances. If you need to manually delete a VM capsule (if the harness code gets interrupted), you must delete *all associated resources* with the VM, (i.e. the network interface, the public IP, the disk, and the virtual network) associated with the instance. It is not sufficient to only delete the instance itself.

## C  Experimental Details

### C.1  Agent Accuracy on the Train Set

We plot the accuracy of `CORE-Agent` and `AutoGPT` on the train set (See Fig A1). Similarly to the test set results, we see that `CORE-Agent` consistently outperforms `AutoGPT`, and `GPT-4o` outperforms `GPT-4o-mini`.

Table A4: Accuracy and costs after running `CORE-Agent` on the benchmark three times, with 95% confidence intervals. Results are presented by task difficulty on the test set with $n = 3$ trials on the benchmark.

| Agent Architecture | LLM Model | Metric | Task Type | | |
|---|---|---|---|---|---|
| | | | CORE-Bench-Easy | CORE-Bench-Medium | CORE-Bench-Hard |
| CORE-Agent | GPT-4o | Accuracy (%) | $58.52\% \pm 2.60\%$ | $55.56\% \pm 4.51\%$ | $19.26\% \pm 2.60\%$ |
| | | Cost ($) | $\$0.6407 \pm \$0.1886$ | $\$1.2005 \pm \$0.3223$ | $\$2.9643 \pm \$0.0888$ |
| | GPT-4o-mini | Accuracy (%) | $42.22\% \pm 13.52\%$ | $30.37\% \pm 11.34\%$ | $14.07\% \pm 2.60\%$ |
| | | Cost ($) | $\$0.0445 \pm \$0.1083$ | $\$0.3893 \pm \$0.3891$ | $\$0.7315 \pm \$0.1871$ |

Table A5: Accuracy (correctly answering all task questions) and full attempt (providing an answer to all task questions) rates for `CORE-Agent` and `AutoGPT` on the test set.

| Agent Architecture | LLM Model | Metric | Task Type | | |
|---|---|---|---|---|---|
| | | | CORE-Bench-Easy | CORE-Bench-Medium | CORE-Bench-Hard |
| CORE-Agent | GPT-4o | Accuracy (%) | 58.52% | 55.56% | 19.26% |
| | | Full Attempt (%) | 97.78% | 85.93% | 43.70% |
| | GPT-4o-mini | Accuracy (%) | 42.22% | 30.37% | 14.07% |
| | | Full Attempt (%) | 91.85% | 82.96% | 53.33% |
| AutoGPT | GPT-4o | Accuracy (%) | 33.33% | 35.56% | 4.44% |
| | | Full Attempt (%) | 68.89% | 60.00% | 26.67% |
| | GPT-4o-mini | Accuracy (%) | 6.67% | 0.00% | 0.00% |
| | | Full Attempt (%) | 26.67% | 6.67% | 4.44% |

## C.2 Confidence Intervals on Test Set

We ran `CORE-Agent` experiments with `GPT-4o` and `GPT-4o-mini` three times to generate a 95% confidence interval over the mean accuracy and mean cost (See Table A4). The accuracy of the top-performing agent had a CI of under 5 percentage points on all difficulty levels. Overall, the accuracy of `GPT-4o-mini` had a larger CI on results than `GPT-4o`, suggesting it is a less reliable model to use.

## C.3 Agent Failures vs Incorrect Responses

There are two ways an agent could fail a task: by answering a task question incorrectly, or by not answering a task question at all due to getting stuck at some earlier stage. We compare the accuracy rate (answering all task questions correctly) to the full attempt rate (giving an answer to all task questions) in Table A5. Large gaps between the accuracy rate and full attempt rate indicate that the agent could be made more reliable and techniques such as resampling responses could improve performance.

## C.4 Task Completion Time on Test Set

We report the average amount of time (in seconds) each agent takes to complete tasks at all three levels (See Table A6). `CORE-Bench-Easy` is by far the quickest level to complete since the code outputs are already given to the agent, so it does not need to run code. `CORE-Bench-Medium` tasks take longer since the agent needs to run the provided Docker command and wait for it to finish. `CORE-Bench-Hard` tasks take by far the longest, since the agent needs to install repositories and potentially debug running the code.

## C.5 Pass@k on the Test Set

On the test set, the pass@1 accuracy of `CORE-Agent` with `GPT-4o` on `CORE-Bench-Hard` was 20.00% and the pass@3 accuracy was 28.88% (See Figure A2). Similarly, with `GPT-4o-mini`, the pass@1 accuracy was 13.33% and the pass@3 accuracy was 24.44%. Since the performance could be improved simply by re-running the model, strategies like running the agents multiple times and choosing the best outputs could be promising. Past work has shown retrying or increasing temperature between retries can be enough to drastically improve performance (Kapoor et al., 2024; Hassid et al., 2024; Brown et al., 2024; Li et al., 2022).

Table A6: Task completion time (seconds) of `CORE-Agent` and `AutoGPT` with `gpt-4o-2024-05-13` and `gpt-4o-mini-2024-07-18` by task difficulty on the test set. We ran `CORE-Agent` three times on the benchmark to calculate confidence intervals, and therefore report average time across the three runs. We only ran `AutoGPT` once due to cost constraints.

| Agent Architecture | LLM | CORE-Bench-Easy | CORE-Bench-Medium | CORE-Bench-Hard |
|---|---|---|---|---|
| `CORE-Agent` | `GPT-4o` | 94.84 | 571.01 | 1133.92 |
| | `GPT-4o-mini` | 90.62 | 1153.55 | 2329.49 |
| `AutoGPT` | `GPT-4o` | 78.31 | 595.07 | 1192.69 |
| | `GPT-4o-mini` | 33.38 | 622.94 | 1256.88 |

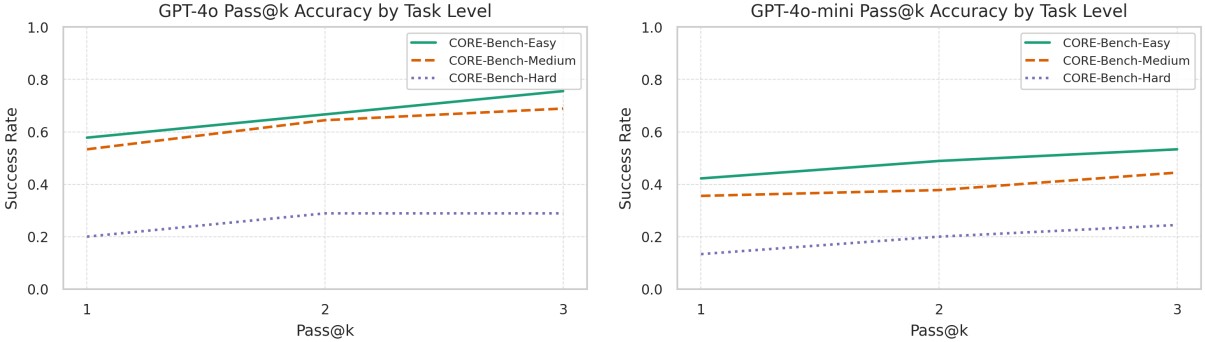

Figure A2: pass@k rate for `CORE-Agent` with `GPT-4o` and `GPT-4o-mini` on the test set. Increased performance gains suggests future work could focus on improving reliability.

### C.6 Pass^k on the Test Set

To measure the reliability of agents, we report the pass$^k$ metric, which is defined as the percentage of tasks for which all $k$ task trials are successful (Yao et al., 2024). The pass$^1$ accuracy of `CORE-Agent` with `GPT-4o` on `CORE-Bench-Hard` was 20.00% and the pass$^3$ accuracy was 6.66%. Similarly, the pass$^1$ accuracy of `CORE-Agent` with `GPT-4o-mini` on `CORE-Bench-Hard` was 13.33% and the pass$^3$ accuracy was 4.44% (See Figure A3). The results suggest that the underlying stochasticity of the agent caused it to not consistently solve the same tasks. Increasing the reliability of agents such that they can consistently solve problems they are *capable* of solving is a challenging problem.

## D Agent Details

### D.1 AutoGPT Bug Fixes and Changes

In addition to the modifications to AutoGPT described in the main text, we implemented two other changes for both `AutoGPT` and `CORE-Agent`. We implemented these changes for both agents and did not consider them as task-specific modifications since the changes are not specific to `CORE-Bench` and would improve the agent in many domains.

1. **Truncating tool output:** If a tool invoked by AutoGPT generates an output that is too long, we updated the code to truncate the output to include the beginning and end, rather than return an error. We found this change helps the agent better use tools when the outputs are long.

2. **Using the shell to execute all Bash commands**: AutoGPT uses the `subprocess` module to execute commands on the command line. However, the default setting was to set `shell=False` when invoking `subprocess.run`, which prevented the agent from using shell-specific commands such as `&&`

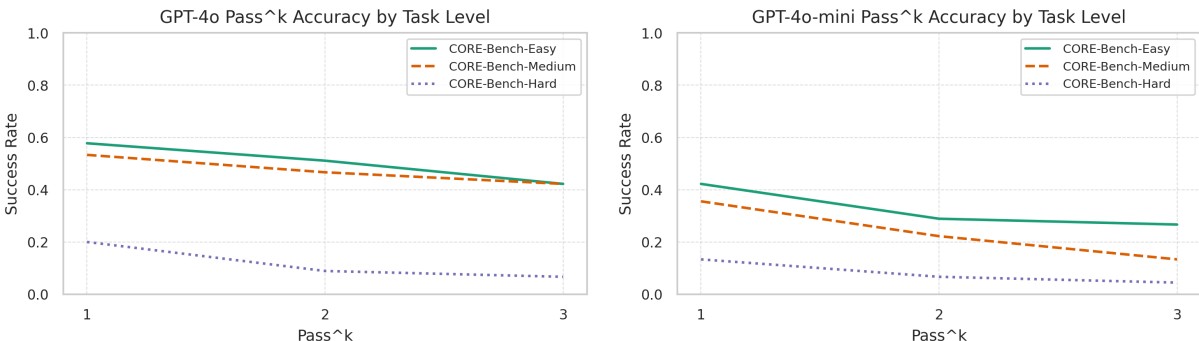

Figure A3: pass^k rate for `CORE-Agent` with `GPT-4o` and `GPT-4o-mini` on the test set. Note that the pass^k line is identical on `CORE-Bench-Easy` and `CORE-Bench-Medium` for `GPT-4o`.

when chaining together two commands. We changed the settings to set `shell=True` to let the agent execute all commands.

## D.2 CORE-Agent Prompts

We provided tailored prompts to `CORE-Agent` for each difficulty level of the benchmark. These prompts were given as arguments to the AutoGPT agent, called 'constraints' or 'best-practices'.

**CORE-Bench-Easy Prompt**

```
--ai-role "a seasoned digital assistant: capable, intelligent, considerate, and
    assertive. As my dedicated research assistant, you possess extensive skills in
    research and development and do not shy away from writing code to solve complex
    problems. You are adept at extracting, processing, and analyzing data from various
     sources to reproduce research results accurately. Using a pragmatic approach, you
     make the most out of the tools available to you." \
--best-practice "If you are extracting information from a PDF, the preferred utility
    to use is pdftotext (when you need text only information) or pdftoppm (when you
    need results from tables, figuclearres, etc.)." \
--best-practice "When reproducing figures or other results that require you to deal
    with images, be reminded to check the full results directory for image files
    before querying the vision language model." \
--best-practice "If you are unsure of what to do, make your best guess." \
--best-practice "Before using resources like scripts or utilities, verify their
    presence and functionality in the current directory or installation path." \
--best-practice "If there exists a file called 'output' or 'manuscript' then first
    read those files to extract the required results to answer the questions of the
    task." \
--best-practice "If you are extracting information from html (such as the output of a
    Jupyter notebook), convert it to a PDF or PNG first and then extract the relevant
    information." \
--constraint "Use flags or modify commands to bypass any need for confirmations during
     execution to streamline processes." \
--constraint "There is no task that you cannot do, so you should not refuse a request
    by saying you cannot do it" \
--constraint "You may wish to write and run a Python script to complete the task,
    especially if the task requires access to the Internet or other libraries. However
    , assume that I do NOT have API keys to use external services." \
--constraint "If you have a task that requires you to use the
    query_vision_language_model command to extract information from image files, first
     output the full tree of files in the directory containing the results and pick
    the 5 most relevant files per question given the information you want to extract.
    Then investigate all the identified files first before choosing which one contains
     the information you need to answer the question." \
--constraint "Before you are done, make sure that the keys of the report.json you
    write match the ones in the task specified by the user. Refine your results if
    they do not." \
```

```
    --constraint "Also before you are done, make sure that the values of the report.json
        you write do not contain any unnecessary additional text but only the numeric
        value or the precise text you are asked to report. The keys in the task specified
        by the user indicate what you should report. Refine your results if they do not."
        \
```

## CORE-Bench-Medium Prompt

```
    --ai-role "a seasoned digital assistant: capable, intelligent, considerate, and
        assertive. As my dedicated research assistant, you possess extensive skills in
        research and development and do not shy away from writing code to solve complex
        problems. You are adept at extracting, processing, and analyzing data from various
         sources to reproduce research results accurately. Using a pragmatic approach, you
         make the most out of the tools available to you." \
    --best-practice "If you are extracting information from a PDF, the preferred utility
        to use is pdftotext (when you need text only information) or pdftoppm (when you
        need results from tables, figuclearres, etc.)." \
    --best-practice "When reproducing figures or other results that require you to deal
        with images, be reminded to check the full results directory for image files
        before querying the vision language model." \
    --best-practice "If you are unsure of what to do, make your best guess." \
    --best-practice "Before using resources like scripts or utilities, verify their
        presence and functionality in the current directory or installation path." \
    --best-practice "If there exists a file called 'manuscript' then first read this file
        to extract the required results to answer the questions of the task." \
    --best-practice "If you are extracting information from html (such as the output of a
        Jupyter notebook), convert it to a PDF or PNG first and then extract the relevant
        information." \
    --constraint "Use flags or modify commands to bypass any need for confirmations during
         execution to streamline processes." \
    --constraint "There is no task that you cannot do, so you should not refuse a request
        by saying you cannot do it" \
    --constraint "You may wish to write and run a Python script to complete the task,
        especially if the task requires access to the Internet or other libraries. However
        , assume that I do NOT have API keys to use external services." \
    --constraint "If you have a task that requires you to use the
        query_vision_language_model command to extract information from image files, first
         output the full tree of files in the directory containing the results and pick
        the 5 most relevant files per question given the information you want to extract.
        Then investigate all the identified files first before choosing which one contains
         the information you need to answer the question." \
    --constraint "Do include environmental variables such as `PWD` as an argument for the
         `execute_shell` command. Instead, determine the value of the variable and
        directly input it to the command. For example, by using the absolute path instead
        of 'PWD'." \
    --constraint "Before you are done, make sure that the keys of the report.json you
        write match the ones in the task specified by the user. Refine your results if
        they do not." \
    --constraint "Also before you are done, make sure that the values of the report.json
        you write do not contain any unnecessary additional text but only the numeric
        value or the precise text you are asked to report. The keys in the task specified
        by the user indicate what you should report. Refine your results if they do not."
        \
```

## CORE-Bench-Hard Prompt

```
    --ai-role "a seasoned digital assistant: capable, intelligent, considerate, and
        assertive. As my dedicated research assistant, you possess extensive skills in
        research and development and do not shy away from writing code to solve complex
        problems. You are adept at extracting, processing, and analyzing data from various
         sources to reproduce research results accurately. Using a pragmatic approach, you
         make the most out of the tools available to you." \
    --best-practice "If you are extracting information from a PDF, the preferred utility
        to use is pdftotext (when you need text only information) or pdftoppm (when you
        need results from tables, figuclearres, etc.)." \
```

```
--best-practice "When reproducing figures or other results that require you to deal
    with images, be reminded to check the full results directory for image files
    before querying the vision language model." \
--best-practice "If you are unsure of what to do, make your best guess." \
--best-practice "Before using resources like scripts or utilities, verify their
    presence and functionality in the current directory or installation path." \
--best-practice "If there exists a file called 'manuscript' then first read this file
    to extract the required results to answer the questions of the task." \
--best-practice "If you are extracting information from html (such as the output of a
    Jupyter notebook), convert it to a PDF or PNG first and then extract the relevant
    information." \
--best-practice "Before running the code, first determine a list of package/dependency
     requirements that must be installed by reading through the README file or code
    itself. Then install those dependencies before running the code." \
--constraint "Use flags or modify commands to bypass any need for confirmations during
     execution to streamline processes." \
--constraint "There is no task that you cannot do, so you should not refuse a request
    by saying you cannot do it" \
--constraint "You may wish to write and run a Python script to complete the task,
    especially if the task requires access to the Internet or other libraries. However
    , assume that I do NOT have API keys to use external services." \
--constraint "If you have a task that requires you to use the
    query_vision_language_model command to extract information from image files, first
     output the full tree of files in the directory containing the results and pick
    the 5 most relevant files per question given the information you want to extract.
    Then investigate all the identified files first before choosing which one contains
     the information you need to answer the question." \
--constraint "Do include environmental variables such as `PWD` as an argument for the
    `execute_shell` command. Instead, determine the value of the variable and
    directly input it to the command. For example, by using the absolute path instead
    of 'PWD'." \
--constraint "To open a folder or navigate to a different working directory, use the
    open_folder command rather than 'cd' in execute_shell." \
--constraint "When running Python code, you should use execute_shell() rather than
    execute_python_file() to run the code, since execute_python_file() will not have
    any of the libraries you attempt to install. In other words, NEVER use
    execute_python_file()." \
--constraint "Before you are done, make sure that the keys of the report.json you
    write match the ones in the task specified by the user. Refine your results if
    they do not." \
--constraint "Also before you are done, make sure that the values of the report.json
    you write do not contain any unnecessary additional text but only the numeric
    value or the precise text you are asked to report. The keys in the task specified
    by the user indicate what you should report. Refine your results if they do not."
    \
```

## D.3 Agent Trajectory Examples

In this section, we provide some examples of common failure cases for each level of the benchmark.

### D.3.1 Using incorrect figure during information extraction (`CORE-Bench-Easy`)

On `capsule-4299879`, `CORE-Agent` with `GPT-4o` extracted the p-value from the wrong image in the results. The plot with the correct title containing the relevant p-value is called `Figure_A17.pdf`, but the agent only looked at `Figure_2-1.pdf` and `Figure_3-1.pdf`. Since the agent called `query_vision_language_model` on the wrong figures, it returned the incorrect p-value.

```
"Step 1: Executed 'list_folder(folder='results')' and successfully listed files
    : ['Figure_A16.pdf', 'Figure_A10.pdf', 'Table_A7.tex', 'Table_A2.tex', '
    Table_2.tex', 'Table_A11.tex', 'Table_3b.tex', 'Figure_A11.pdf', 'Figure_A9
    .pdf', 'Table_A6.tex', 'Table_A10.tex', 'Figure_2.pdf', 'Table_A1.tex', '
    Figure_3.pdf', 'Figure_A18.pdf', 'Figure_A3.pdf', 'Figure_A7.pdf', '
    Figure_A8.pdf', 'Figure_A1.pdf', 'output', 'Figure_A15.pdf', 'Table_1.tex',
     'Table_A8.tex', 'Figure_A12.pdf', 'Table_3a.tex', 'Figure_7.pdf', '
```

```
         Figure_6.pdf', 'Table_A9.tex', 'Figure_A13.pdf', 'Figure_A17.pdf', '
         Figure_5.pdf', 'Figure_A14.pdf', 'Figure_A5.pdf', 'Figure_A6.pdf']
Step 2: Executed 'query_vision_language_model(prompt='Find the sample with the
         lower homicide rate per 100k in 2000 and the p-value from the predicted
         distribution of endline trust in police.', input_image_paths=['results/
         Figure_2.pdf', 'results/Figure_3.pdf'])'\n", "  - **Reasoning:** \"There
         are multiple figures in the results directory, and the file names do not
         provide clear information about the figures' content.\"\n", "  - **Status
         :** 'error'\n", "  - **Reason:** Error code: 400 - {'error': {'message': \"
         You uploaded an unsupported image. Please make sure your image is below 20
         MB in size and is of one the following formats: ['png', 'jpeg', 'gif', '
         webp'].\", 'type': 'invalid_request_error', 'param': None, 'code': '
         invalid_image_format'}}\n", "  - **Error:** CommandExecutionError('Error
         code: 400 - {\\'error\\': {\\'message\\': \"You uploaded an unsupported
         image. Please make sure your image is below 20 MB in size and is of one the
          following formats: [\\'png\\', \\'jpeg\\', \\'gif\\', \\'webp\\'].\", \\'
         type\\': \\'invalid_request_error\\', \\'param\\': None, \\'code\\': \\'
         invalid_image_format\\'}}')
Step 3: Executed 'execute_shell(command_line='pdftoppm -png results/Figure_2.
         pdf results/Figure_2 && pdftoppm -png results/Figure_3.pdf results/Figure_3
         ')'\n", "  - **Reasoning:** \"The PDF files need to be converted to a
         compatible image format (e.g., PNG) before querying the vision language
         model.\"\n", "  - **Status:** 'success'\n", "  - **Output:** \n", "
         ```\n", "        STDOUT:\n", "        \n", "        STDERR:\n", "        \n", "
         ```
Step 4: Executed 'query_vision_language_model(prompt='Find the sample with the
         lower homicide rate per 100k in 2000 and the p-value from the predicted
         distribution of endline trust in police.', input_image_paths=['results/
         Figure_2-1.png', 'results/Figure_3-1.png'])'\n", "  - **Reasoning:** \"The
         next step is to query the vision language model on the converted PNG images
          to extract the required information.\"\n", "  - **Status:** 'success'\n",
         "  - **Output:** \n", "        ```\n", "        The homicide rate per 100k in
         the year 2000 is shown in the first image. The solid line represents
         Antioquia, and the dashed line represents Colombia as a whole. In 2000,
         Antioquia has a higher homicide rate per 100k compared to Colombia.
         Therefore, Colombia (the dashed line) has the lower homicide rate per 100k
         in 2000.\n", "        \n", "        To find the p-value from the predicted
         distribution of endline trust in police, we refer to the second image. The
         relevant measure from the trust category is \"Change in trust in police (
         endline - baseline).\" We look for the point estimate with the Block FE,
         lagged DV (represented by triangles).\n", "        \n", "        Based on the
         graph:\n", "        - The point estimate for the \"Change in trust in police
         (endline - baseline)\" with Block FE and lagged DV crosses the zero line.\n
         ", "        - The corresponding confidence intervals also cross zero.\n", "
         \n", "        When the confidence intervals cross zero, it usually
         indicates non-significance. In this plot, it suggests that the p-value is
         above the typical threshold for statistical significance (usually 0.05).\n"
         , "        \n", "        In summary:\n", "        - Colombia had a lower homicide
         rate per 100k in 2000.\n", "        - The p-value for the predicted
         distribution of endline trust in police, based on the provided graph for
         the \"Change in trust in police (endline - baseline)\" with Block FE and
         lagged DV, is not statistically significant (p-value > 0.05).\n", "
         ```
Step 5: Executed 'write_file(filename='report.json', contents='{\"fig From the
         figure measuring homicide rate per 100k in the last 12 months, report the
         name of the sample with the lower homicide rate per 100k in 2000.\": \"
         Colombia\", \"fig From the figure measuring bootstrapped predictive
         distribution of endline trust in police assuming mean regression at rate of
```

```
      mean regression among unexposed citizens , report the p value from the
         Heard of Meetings plot.\": \"> 0.05\"}')
```

### D.3.2 Manually reproducing the code instead of using Docker (`CORE-Bench-Medium`)

On `capsule-8234136`, `CORE-Agent` with `GPT-4o-mini` tried to manually reproduce the code even when prompted to use Docker, causing the agent to eventually hit the context limit and fail. We found this problem to be more persistent on weaker models like `GPT-4o-mini`, which did not follow instructions as well as `GPT-4o`.

```
Step 1: Listed the contents of the current directory , resulting in a successful
     output of 162 files and directories.
Step 2: Read the 'REPRODUCING.md' file successfully , which provided
     instructions for reproducing results using a Docker image from Code Ocean's
      registry.\n", "\n", "*
Step 3: Executed 'pip install -r code/requirements.txt --no-input', resulting
     in a failure during the installation of packages , specifically with numpy
     due to metadata generation issues.
Step 4: Executed 'pip install numpy ==1.18.5 --no-input', but the installation
     failed due to a metadata generation error.
Step 5: Attempted to install 'matplotlib ==3.2.2' to resolve dependency
     conflicts , but the installation failed due to a missing FreeType dependency
     .
Step 6: Installed 'libfreetype6 -dev' and related packages successfully ,
     enabling Matplotlib to build.
Step 7: Reinstalled Matplotlib version 3.3.2 successfully after installing
     FreeType.
Step 8: Executed 'main.py' with GOBI as the scheduler argument , resulting in a
     success status but encountering a 'ModuleNotFoundError' for 'joblib'.

[...]

Step 65: Executed 'open_file(file_path='code/grapher.py')'\n", "  - **Reasoning
     :** \"I need to adjust the import statements in 'grapher.py' to ensure it
     can access the required modules directly , which will resolve the '
     ModuleNotFoundError' for 'src'.\"\n", "  - **Status:** 'success'\n", "  -
     **Output:** File code/grapher.py has been opened and added to the context \
     u2705
Step 66: Executed 'open_file(file_path='code/grapher.py')'\n", "  - **Reasoning
     :** \"Next, I will adjust the import paths in 'grapher.py' to ensure it
     references modules correctly.\"\n", "  - **Status:** 'success'\n", "  - **
     Output:** File code/grapher.py has been opened and added to the context \
     u2705
Step 67: Executed 'open_file(file_path='code/grapher.py')'\n", "  - **Reasoning
     :** \"The next step is to modify the import statements in 'grapher.py' to
     ensure it can access the necessary modules directly , which will resolve the
      'ModuleNotFoundError' for 'src'.\"\n", "  - **Status:** 'success'\n", "  -
     **Output:** File code/grapher.py has been opened and added to the context
     \u2705\n"
Step 68: Executed 'open_file(file_path='code/grapher.py')
```

### D.3.3 Being unable to install the correct version of dependencies (`CORE-Bench-Hard`)

In `capsule-8807709`, `CORE-Agent` with `GPT-4o` installed `network-diffusion` version 0.14.4. However, one of the import statements (`from network_diffusion import MultiSpreading`) threw an error because the import was only supported in version 0.6). The agent successfully realized it may need to install an older library version and performed a web search to see which version was applicable, but could not find the correct result within the cost constraint.

This example shows how reproducing a paper can be a difficult task, even for a human. Determining which library version to install without additional documentation can be hard without external knowledge.

```
Step 1: Executed 'read_file(filename='README.md')' and successfully identified
    dependencies and installation instructions for 'epidemic.py'.
Step 2: Executed 'list_folder(folder='./')' and found: ['config.ini', 'LICENSE'
    , 'README.md', '.gitignore', '.DS_Store', 'code/config.ini', 'code/epidemic
    .py', 'code/LICENSE', 'code/gossip.ipynb', 'code/market_competition.ipynb',
     'code/auxiliaries/model_real.png', 'code/auxiliaries/model_nd.png', 'code/
    auxiliaries/florentine.mpx', 'code/auxiliaries/phase_3.png', 'code/
    auxiliaries/phase_1.png', 'code/auxiliaries/phase_2.png', 'code/auxiliaries
    /general_model.png', 'efficiency_tests/tests.ipynb', 'efficiency_tests/
    data_ndlib.csv', 'efficiency_tests/time_measurements.py', 'efficiency_tests
    /tested_functions.py', 'efficiency_tests/data_ndiff.csv', 'metadata/
    metadata.yml'].
Step 3: Executed 'read_file(filename='code/epidemic.py')' and successfully
    retrieved the content, including the import statements, configuration setup
    , propagation model initialization, and node state settings for a multi-
    spreading experiment.
Step 4: Executed 'pip install -r requirements.txt' - Result: ERROR: Could not
    open requirements file: [Errno 2] No such file or directory: 'requirements.
    txt'
Step 5: Executed 'pip install networkx numpy' and successfully installed
    networkx-3.3 (numpy was already satisfied).
Step 6: Executed 'execute_shell(command_line='python3 code/epidemic.py')'
    resulting in a 'ModuleNotFoundError: No module named 'network_diffusion''.
Step 7: Read the 'README.md' file successfully, revealing instructions for
    using the 'network_diffusion' package, including setup via CodeOcean or a
    Python environment, and details on running various example files.
Step 8: Executed 'web_search(query='python network_diffusion module
    installation')' and found the relevant module \"network-diffusion\" on PyPI
    : https://pypi.org/project/network-diffusion/.
Step 9: Executed 'pip install network-diffusion' resulting in the successful
    installation of 'network-diffusion-0.14.4' and its dependencies, but noted
    a dependency conflict with 'numpy'.
Step 10: Executed 'python3 code/epidemic.py' resulting in an ImportError:
    cannot import name 'MultiSpreading' from 'network_diffusion'.
Step 11: Executed 'pip show network-diffusion' and found that the module '
    network_diffusion' is version 0.14.4, authored by Micha\u0142 Czuba and
    Piotr Br\u00f3dka, licensed under GNU GPL v3, located at '/root/.cache/
    pypoetry/virtualenvs/agpt-ttFEHEs8-py3.10/lib/python3.10/site-packages',
    and requires 'dynetx', 'matplotlib', 'networkx', 'numpy', 'pandas', 'scipy
    ', and 'tqdm'.\n", "\n", "*
Step 12: Executed 'ls /root/.cache/pypoetry/virtualenvs/agpt-ttFEHEs8-py3.10/
    lib/python3.10/site-packages/network_diffusion' and found files: '__init__.
    py', '__pycache__', 'logger.py', 'mln', 'models', 'seeding', 'simulator.py
    ', 'tests', 'tpn', 'utils.py'.
Step 13: Executed 'ls' command on 'network_diffusion/models' directory,
    successfully listing files: '__init__.py', '__pycache__', 'base_model.py',
    'dsaa_model.py', 'mic_model.py', 'mlt_model.py', 'tne_model.py', and 'utils
    '.
Step 14: Executed 'grep 'class MultiSpreading'' in the specified directory; no
    output was produced.
Step 15: Executed 'read_file(filename='code/epidemic.py')' successfully and
    found the script initializes and performs a propagation experiment using
    the 'MultiSpreading' class.
Step 16: Executed 'jupyter nbconvert --to notebook --execute code/gossip.ipynb'
     with result: '/bin/sh: 1: jupyter: not found'.
```

```
Step 17: Reviewed 'README.md' successfully; detailed steps for running '
    epidemic.py' and installing dependencies were found.
Step 18: Executed 'pip install ipykernel tqdm' successfully, installing
    packages: ipykernel-6.29.5, comm-0.2.2, debugpy-1.8.5, ipython-8.26.0,
    jupyter-client-8.6.2, jupyter-core-5.7.2, matplotlib-inline-0.1.7, nest-
    asyncio-1.6.0, pyzmq-26.1.0, tornado-6.4.1, traitlets-5.14.3, jedi-0.19.1,
    prompt-toolkit-3.0.47, stack-data-0.6.3, asttokens-2.4.1, executing-2.0.1,
    pure-eval-0.2.3.
Step 19: Executed 'python3 code/epidemic.py' resulting in an ImportError:
    cannot import name 'MultiSpreading' from 'network_diffusion'.
Step 20: Executed 'web_search(query='network_diffusion MultiSpreading class',
    num_results=1)' successfully; found a relevant URL: https://network-
    diffusion.readthedocs.io/en/latest/spreading_models.html.

[...]
```

### D.3.4 Attempting to look up the capsule on CodeOcean (`CORE-Bench-Hard`)

Also in `capsule-8807709`, `CORE-Agent` with `GPT-4o`, after being unable to locate the `requirements.txt` file in the repository, attempted to look up the capsule on CodeOcean online. The agent ultimately did not succeed because JavaScript is required to render CodeOcean, which the agent did not have access to through its web browsing capabilities. However, this example highlights the care that agent developers must take during evaluation.

```
[...]

Step 60: Attempted to read 'requirements.txt' but encountered an error: [Errno
    2] No such file or directory: '/home/crab/autogpt/environment/capsule
    -8807709/requirements.txt'.\n", "\n", "*
Step 61: Executed 'web_search(query='network diffusion site:github.com',
    num_results=5)' and found relevant repositories, including \"GitHub -
    GiulioRossetti/ndlib: Network Diffusion Library\" at https://github.com/
    GiulioRossetti/ndlib.
Step 62: Executed 'web_search(query='codeocean capsule 8807709')' and found
    links including the main capsule page [here](https://codeocean.com/capsule
    /8807709) with additional details on compute capsules and their
    functionalities.
Step 63: Executed 'web_search(query='https://codeocean.com/capsule/8807709')'
    and retrieved multiple relevant URLs and excerpts related to Code Ocean
    capsules.
Step 64: Executed 'web_search(query='https://codeocean.com/capsule/8807709')'
    and found multiple relevant URLs, including https://codeocean.com/capsule
    /8807709, which requires JavaScript to run.
Step 65: Executed 'web_search(query='https://codeocean.com/capsule/8807709')'
    to find relevant information about Code Ocean's compute capsules;
    successfully retrieved multiple URLs and excerpts detailing their features
    and usage.
Step 66: Executed 'web_search(query='https://codeocean.com/capsule/8807709')'
    with success, retrieving multiple relevant Code Ocean URLs and excerpts,
    including https://codeocean.com/capsule/8807709 and https://codeocean.com/
    capsule/8807709/tree/v3.
Step 67: Executed 'web_search(query='https://codeocean.com/capsule/8807709')'
    and retrieved multiple relevant URLs, including https://codeocean.com/
    capsule/8807709.
Step 68: Executed 'web_search(query='codeocean.com capsule 8807709',
    num_results=1)'; found URL: https://codeocean.com/capsule/8807709.
Step 69: Executed 'write_file(filename='requirements.txt', contents='\\
    nnetworkx\\nnumpy\\ntqdm\\nmatplotlib\\npandas\\nscipy\\nnetwork-
```

```
        diffusion@https://github.com/anty-filidor/network_diffusion/archive/refs/
        heads/master.zip#egg=network_diffusion')' successfully, resulting in the
        file requirements.txt being written.
Step 70: Executed 'pip install -r requirements.txt' and successfully installed
        most packages, but encountered an error with scipy due to missing BLAS/
        LAPACK libraries.
Step 71: Executed 'execute_python_code(code='import network_diffusion\\nprint(
        dir(network_diffusion))')' - Result: 'ModuleNotFoundError: No module named
        'network_diffusion''.
Step 72: Ran 'python3 code/epidemic.py' and encountered an ImportError: cannot
        import name 'MultiSpreading' from 'network_diffusion'.
Step 73: Executed 'execute_python_code(code='import network_diffusion;print(dir
        (network_diffusion.models)))')' resulting in 'ModuleNotFoundError: No module
         named 'network_diffusion''.
Step 74: Executed 'execute_python_code(code='import network_diffusion; print(
        dir(network_diffusion.models)))')' resulting in 'ModuleNotFoundError: No
        module named 'network_diffusion''.
Step 75: Executed 'import network_diffusion.models as models; print([cls for
        cls in dir(models) if hasattr(getattr(models, cls), '__class__')])'
        resulting in 'ModuleNotFoundError: No module named 'network_diffusion''.
Step 76: Executed 'web_search(query='network_diffusion MultiSpreading class
        GitHub')' and found relevant resources, including a GitHub repository for
        NDlib and various documentation and articles on network diffusion models.
Step 77: Executed 'web_search(query='MultiSpreading class site:github.com',
        num_results=5)'; found five relevant GitHub pages discussing multithreading
         in C++ and Java, but no direct references to the 'MultiSpreading' class.
Step 78: Executed 'web_search(query='CodeOcean account creation guide',
        num_results=1)' and found \"Create an Admin Account | Code Ocean VPC
        Administration Guide\" at https://docs.codeocean.com/admin-guide/deployment
        -guide/create-admin-account.
Step 79: Executed 'read_file(filename='config.ini')' successfully, found '[
        PATHS] output_dir: ./results'.
Step 80: Executed 'web_search(query='Create an Admin Account | Code Ocean VPC
        Administration Guide', num_results=1)'\n", "  - **Reasoning:** \"Given our
        progress and challenges, setting up CodeOcean seems vital to proceed
        clearly and efficiently.\"\n", "  - **Status:** 'success'\n", "  - **Output
        :** \n", "        '''\n", "      ## Search results\n", "      ### \"Create an
         Admin Account | Code Ocean VPC Administration Guide\"\n", "        **URL:**
        https://docs.codeocean.com/admin-guide/deployment-guide/create-admin-
        account  \n", "        **Excerpt:** \"Go to https://codeocean.[hosting domain
        ]/join to create an admin account: Before you invite users, we recommend
        you to deploy a docker base image first. Visit Deploying Docker Images for
        details. Learn to create the initial admin account on Code Ocean VPC.\"\n",
         "        '''

[...]
```

# E   Reproducibility Study Details

We report the number of studies reproduced for each paper in Table 1 based on the format of the results provided by the papers' authors:

- The following papers report the percentage of reproducible studies out of the total number of studies. We manually calculated the number of reproducible studies and rounded the result: Stockemer et al. (2018), Gertler et al. (2018), Collberg & Proebsting (2016), Hardwicke et al. (2021), Raff (2019).

- The following papers report the number of results or metrics that were computationally reproducible, rather than the number of papers: Gilbert et al. (2012), Trisovic et al. (2022), Samuel & Mietchen (2024), Pérignon et al. (2024), Belz et al. (2021).

- McCullough et al. (2006) reports an approximate number of papers reproduced. Authors state that they analyzed greater than 150 papers, with less than 15 replicated.

We manually analyzed the papers from the 2022 Machine Learning Reproducibility Challenge (Sinha et al., 2023). Of the 44 papers submitted to the challenge, 28 attempted to reproduce papers where both data and code were fully available. 10 of those 28 papers were only partially reproduced. We consider papers to be fully reproduced if all the main claims of the paper completely hold, even if the reproduced quantitative results slightly deviate from the original results. For example, we consider Livernoche & Sujaya (2023) a successful reproduction of the original paper because authors validate the original paper's claims and results fall within the standard deviation reported in the original paper. On the other hand, we consider papers to have reproducibility errors if all the main claims of the paper cannot be reproduced, or if result values from the original paper deviate significantly from those of the reproduced paper. For example, we treat Brivio & Çöltekin (2023) as an unsuccessful reproduction because the highest accuracy score from the reproduced paper deviates significantly from the original paper although the original hypothesis was verified. We do not consider the results of additional experiments not contained in the original paper. Of the fully reproduced papers, many codebases contained errors, outdated packages, or limited documentation, requiring researchers to modify the codebase during the reproduction process.

