# OpenReview forum: "CORE-Bench: Fostering the Credibility of Published Research Through a Computational Reproducibility Agent Benchmark"
_TMLR — Accepted by TMLR_

### Review · Reviewer_VqsX · 2024-10-09

**Summary Of Contributions:**

This work introduces CORE-Bench, a benchmark designed to evaluate agents' ability to reproduce the results of scientific studies using provided data and code. The work filtered and selected 90 capsules from CodeOcean, spanning medical science, social science, and computer science, and utilizing Python and R. For each capsule, three task difficulty levels (Easy, Medium, and Hard) are created. The work measures the agent performance with the proportion of tasks in which all questions are answered correctly, defined as results falling within a 95% prediction interval derived from three manual runs. Two baselines, AutoGPT and its task-specific variant CORE-Agent, are tested on this benchmark using GPT-4o and GPT-4o-mini. The results demonstrated expected performance variations across difficulty levels and revealed substantial room for agent improvement in solving computational reproduction tasks.

**Audience:**

Yes

**Claims And Evidence:**

Yes

**Requested Changes:**

- **Q1**: What's the exact proportion of R and Python in each discipline? I think Figure 8 can be further improved since we cannot tell any varying performance from this figure in each discipline or programming language if it is disproportional.

- **Q2**: What's the exact proportion of visual and written questions in each difficulty level? I might have missed something in the paper, but I didn't find the exact number. It would be more convincing if some examples were illustrated.

- **Q3**: The paper reports the experimental results of AutoGPT only with different-sized GPT-4o. I would like to know if other LLMs like Claude and Gemini are tested. What's their performance in different levels of the CORE-Bench, and what are any findings compared to the GPT-4o?

- **Q4**: There are various reasons for a task's failure, such as dependency installation and results retrieval. I am wondering if the authors can provide more details. For instance, in each level of difficulty, what are the reasons that lead to failures, and how many tasks are there for each reason?

- **Q5**: What's the time cost for finishing all the tasks at each difficulty level? What are the possible reasons that may lead to more time costs?

I would be more convinced if the above questions could be explained.

**Strengths And Weaknesses:**

**Strengths**:

- Computational reproducibility is an essential subject in scientific research. Using AI agents to verify the reproducibility of published research is interesting.
- The writing is pretty clear and easy to follow.
- The capsules involve several scientific domains for both Python and R. The paper claimed that CORE-Bench is among the first to include tasks in R.


**Weaknesses**:
- The authors mentioned in 2.1 that including irreproducible papers in the benchmark is unnecessary, which I'm afraid I have to disagree with. In real-life settings, irreproducible papers can be easy to find. Observing how the agents perform with these tasks is also an important aspect of evaluating the agent's performance.

---

> ### Author Response · Authors · 2024-10-30
> **Thank you for your review**
>
> > The authors mentioned in 2.1 that including irreproducible papers in the benchmark is unnecessary, which I'm afraid I have to disagree with. In real-life settings, irreproducible papers can be easy to find. Observing how the agents perform with these tasks is also an important aspect of evaluating the agent's performance.
>
> We acknowledge the lack of clarity about not providing irreproducible papers in our benchmark. In response, we have updated section 2.1 to further explain this decision and why we believe our benchmark consists of real-world tasks. A brief overview of what we clarified is included below.
> - We decompose the task of verifying computational reproducibility into two sub-tasks: code reproducibility and result reproducibility. This paper and benchmark focuses on code reproducibility, or running the code and obtaining the results the capsule is supposed to produce, whether or not those match the results reported in the paper (in fact, 30/90 included papers are not result-reproducible). We focus on code reproducibility since it is by far the more time consuming part for a human.
> - Specifically, the benchmark is used to evaluate if agents can automate code reproducibility, as opposed to verifying if papers are reproducible. If in the future a high performing agent is created, it could also help with verifying that existing papers are reproducible, but we are far off from that point. Under this framing, we believe our benchmark consists of real-world tasks since the only way to assess whether an agent can code reproduce the paper is if the paper is known to be code reproducible.
> - Agents developed for this benchmark could be useful to automate reproductions of other papers —once a paper's results are reproduced, verifying that it matches the results in a paper (results reproducibility) is much easier.
> - We did not include code-irreproducible capsules because agents would just fail since the task questions would be impossible to answer. Indeed, we want all benchmark tasks to be human-verified as solvable that way any failure of reproducing a paper can be attributed to a failure of the agent and to a task being unsolvable.
> - For example, SWE-Bench (https://arxiv.org/abs/2310.06770) followed a similar approach by releasing a human-validated subset called SWE-Bench Verified (https://openai.com/index/introducing-swe-bench-verified/) so that perfect performance is attainable, which provides a stronger signal to developers when designing agents. Just as SWE-Bench-Verified doesn’t contain “negative examples” that are unsolvable, neither does CORE-Bench.
>
> > Q1: What's the exact proportion of R and Python in each discipline? I think Figure 8 can be further improved since we cannot tell any varying performance from this figure in each discipline or programming language if it is disproportional.
>
> Medical Sciences: 10 Python, 15 R; Social Sciences: 4 Python, 24 R; Computer Science: 35 Python; 2 R. Since the per-discipline, per-language subgroups are quite small, the confidence intervals would be quite large. However, based on the feedback, we updated Figure A.1 to include this capsule breakdown by discipline and language.
>
> > Q2: What's the exact proportion of visual and written questions in each difficulty level? I might have missed something in the paper, but I didn't find the exact number. It would be more convincing if some examples were illustrated.
>
> Each difficulty level has the same task questions; the difference is in the setup of the repository given to the agent. In the test set, 26/45 capsules have only vision-based questions, 12/45 capsules have only text-based questions, and 7/45 capsules have both. In the train set, 18/45 capsules have only vision-based questions, 23/45 capsules have only text-based questions, and 4/45 capsules have both. We updated Section 2.2 in the paper with this information and added more examples of task questions to the appendix.
>
> > Q3: The paper reports the experimental results of AutoGPT only with different-sized GPT-4o. I would like to know if other LLMs like Claude and Gemini are tested. What's their performance in different levels of the CORE-Bench, and what are any findings compared to the GPT-4o?
>
> Due to budget constraints, we are not currently able to run additional agents. However, we are releasing with the paper a public leaderboard that allows agent developers to easily submit their own agent architectures with underlying model backends to compare for evaluation (see https://anonymous.4open.science/r/core-bench-E7D6/CORE-Bench%20Leaderboard.png for a preview of the leaderboard).

---

> ### Author Response · Authors · 2024-10-30
>
> > Q4: There are various reasons for a task's failure, such as dependency installation and results retrieval. I am wondering if the authors can provide more details. For instance, in each level of difficulty, what are the reasons that lead to failures, and how many tasks are there for each reason?
>
> Based on the reviewer’s suggestion, we ran a qualitative analysis of the failure cases on all levels of the benchmark for the best-performing agent, and included the results in the updated manuscript. A brief summary:
> - All of the failures by CORE-Agent with GPT-4o on CORE-Bench-Easy were caused by an inability to retrieve the relevant information from the code outputs (40.00% of tasks), since the task setup already provides agents with the code outputs.
> - Although CORE-Bench-Medium is strictly harder than CORE-Bench-Easy, since it requires the extra step of running a Docker command to reproduce the code outputs, in practice, CORE-Agent with GPT-4o did almost as well, and failures (42.22% of tasks) were also due to incorrectly retrieving the relevant information from the code outputs.
> - On CORE-Bench-Hard,
>     - The agent successfully completed 22.22% of all tasks by answering all of the task questions correctly.
>     - On 17.78% of tasks, the agent was able to properly install all dependencies and run the code but was unable to retrieve the relevant information from the results to answer the task questions.
>     - On 15.56% of tasks, the agent successfully installed the dependencies but was unable to run the code properly, usually because it did not run the correct command to reproduce the code or it did not run the command from the proper directory.
>     - On the remaining 44.44% of tasks, the agent was unable to properly install the dependencies usually due to installing the incorrect version of libraries and hitting the cost limit. We included this analysis in the updated manuscript.
>
> > Q5: What's the time cost for finishing all the tasks at each difficulty level? What are the possible reasons that may lead to more time costs?
>
> The table below shows the average time cost for running each (model, agent) pair on all 3 levels of the benchmark (in seconds).
>
> ```
> Agent Architecture   | LLM Model    |     Easy Accuracy | Medium Accuracy |  Hard Accuracy
> ______________________________________________________________________________________________
> CORE-Agent           | GPT-4o       |             94.84 |         571.01 |        1133.92
> CORE-Agent           | GPT-4o-mini  |             90.62 |        1153.55 |        2329.49
> AutoGPT              | GPT-4o       |             78.31 |         595.07 |        1192.69
> AutoGPT              | GPT-4o-mini  |             33.38 |         622.94 |        1256.88
> ```
>
> Tasks on CORE-Bench-Easy require the least time since they don’t require running any code – the results are already provided to the agent. CORE-Bench-Hard takes significantly longer than CORE-Bench-Medium since the agent needs to install the libraries and dependencies in addition to just running the code, which often requires trial and error. We added a table that includes the average running time to the appendix.

---

### Review · Reviewer_t3v5 · 2024-10-11

**Summary Of Contributions:**

The authors propose a benchmark dataset, CORE-Bench, for testing AI agents whose job is to check the reproducibility of scientific publications. The benchmark consists of three levels of task difficulty corresponding to different amounts of information/steps of the task which have already been completed that are supplied to the AI agent. Using the results from testing on their benchmark, they make modifications to an existing baseline (AutoGPT) which improve task performance. They also implement an parallel evaluation framework which drastically reduces the amount of time needed to run all of the tasks.

**Audience:**

Yes

**Claims And Evidence:**

Yes

**Requested Changes:**

1. Please provide code which can be run locally, for free, and in a reasonable amount of time which would allow a reader to test a part of your benchmark.

2. Please provide additional justification for the decision not to include any "negative examples," i.e., papers which are not reproducible, in the benchmark.

3. The claim of the header of Section 4.1 is not exactly correct, as the AutoGPT baseline actually performs slightly better on the medium vs. the easy level. Can the authors provide some explanation for this?

Conditional on the resolution of these issues, I would recommend accepting the paper.

---

After the author discussion period, I believe the authors have adequately addressed my concerns. I recommend accepting the paper.

**Strengths And Weaknesses:**

## Strengths

The paper is clearly written and easy to follow. The design choices for each difficulty level of the benchmark tasks are explained thoroughly and seem like a natural progression of skills to test for in reproducibility agents. The attached code appears to be well-documented and user friendly (though there is a caveat to this discussed below), and the evaluation harness provided by the authors seems helpful in drastically reducing evaluation time (provided the user has access to enough virtual machines to make full use of the harness). Testing the evaluation framework on an existing agent (AutoGPT) and using the resulting error logs to make improvements to said agent (CORE-Agent) is also convincing evidence that CORE-Bench can be helpful in developing better reproducibility agents.

This topic is of interest to TMLR's audience. Tools for verifying the reproducibility of scientific work would be of high value for any academic discipline, but especially for ML/AI as the number of published works has exploded in recent years. This paper is an important step towards automatic reproducibility verification, serving as a meta-verification testbed (i.e., a way for verifying that a reproducibility verifier works).

## Weaknesses

The biggest shortcoming is the barrier to reproducing the results in this paper. None of the provided agents can be run without payment, and running the full pipeline without access to Azure takes many days as the authors themselves state. Given that the focus of this paper is on reproducibility, it seems imperative that the authors provide at least one baseline/test setting which can be reproduced locally and for free, so that their framework can be partially checked with relative ease.

Part of the filtering process when creating CORE-Bench was to select tasks which were in fact reproducible. Another important aspect of reproducibility is verifying when previous work is *not* reproducible. This may be a harder task to quantify, as there is basically only one way for reproducibility to succeed (i.e., the code runs and you get the same numbers that were claimed by the authors), but more ways for it to fail (code doesn't run due to dependencies/bugs, experimental results are different, etc.). The authors discuss this choice briefly at the bottom of pg. 5, but I believe more in-depth discussion is necessary to provide compelling evidence for the claim that the proposed benchmark consists of "real-world computational reproducibility tasks."

---

> ### Author Response · Authors · 2024-10-30
> **Thank you for your review**
>
> We thank the reviewer for their thoughtful review. The reviewer provides valuable feedback which we incorporated into the updated manuscript.
>
> > The biggest shortcoming is the barrier to reproducing the results in this paper. None of the provided agents can be run without payment, and running the full pipeline without access to Azure takes many days as the authors themselves state. Given that the focus of this paper is on reproducibility, it seems imperative that the authors provide at least one baseline/test setting which can be reproduced locally and for free, so that their framework can be partially checked with relative ease.
>
> Based on the reviewer’s feedback, we implemented a local evaluation harness that runs agents in a Docker container so that the harness can easily be run without Azure access (https://anonymous.4open.science/r/core-bench-E7D6/ has updated code). In general, it is difficult to provide a LLM-based agent that can run completely free of charge due to the cost of the underlying model backends. However, we suggest two approaches to verify our framework (and we will update the README to make these clear):
>
> 1. We provide an agent called “docker_agent” which reproduces the code by running the Docker command found in the “reproducing.md” file that each capsule in CORE-Bench-Medium has. This allows the user to verify that the harness is working properly, and that each task of the harness is in fact reproducible.
>
> `python3 main.py --experiment_name docker_test --agent_dir agents/docker_agent --no_gpu --task_limit 2 --benchmark_level codeocean_medium --agent_script agent.sh --verbose`
>
> 2. If a user wants to test an LLM-based agent, they can run CORE-Agent with gpt-4o-mini on CORE-Bench-Easy for a minimal cost proof-of-concept. On average, each task would cost $0.0445 to run, and all the user needs is an OpenAI API key (see instructions in repo readme on how to add API key).
>
> `python3 main.py --experiment_name coreagent_test --agent_dir agents/AutoGPT-CORE --no_gpu --task_limit 2 --benchmark_level codeocean_easy --agent_script coreagent_easy_gpt4o-mini.sh --verbose`

---

> ### Author Response · Authors · 2024-10-30
>
> > Part of the filtering process when creating CORE-Bench was to select tasks which were in fact reproducible. Another important aspect of reproducibility is verifying when previous work is not reproducible. This may be a harder task to quantify, as there is basically only one way for reproducibility to succeed (i.e., the code runs and you get the same numbers that were claimed by the authors), but more ways for it to fail (code doesn't run due to dependencies/bugs, experimental results are different, etc.). The authors discuss this choice briefly at the bottom of pg. 5, but I believe more in-depth discussion is necessary to provide compelling evidence for the claim that the proposed benchmark consists of "real-world computational reproducibility tasks."
>
> We acknowledge the lack of clarity about not providing irreproducible papers in our benchmark. In response, we have updated section 2.1 to further explain this decision and why we believe our benchmark consists of real-world tasks. A brief overview of what we clarified is included below.
> - We decompose the task of verifying computational reproducibility into two sub-tasks: code reproducibility and result reproducibility. This paper and benchmark focuses on code reproducibility, or running the code and obtaining the results the capsule is supposed to produce, whether or not those match the results reported in the paper (in fact, 30/90 included papers are not result-reproducible). We focus on code reproducibility since it is by far the more time consuming part for a human.
> - Specifically, the benchmark is used to evaluate if agents can automate code reproducibility, as opposed to verifying if papers are reproducible. If in the future a high performing agent is created, it could also help with verifying that existing papers are reproducible, but we are far off from that point. Under this framing, we believe our benchmark consists of real-world tasks since the only way to assess whether an agent can code reproduce the paper is if the paper is known to be code reproducible.
> - Agents developed for this benchmark could be useful to automate reproductions of other papers —once a paper's results are reproduced, verifying that it matches the results in a paper (results reproducibility) is much easier.
> - We did not include code-irreproducible capsules because agents would just fail since the task questions would be impossible to answer. Indeed, we want all benchmark tasks to be human-verified as solvable that way any failure of reproducing a paper can be attributed to a failure of the agent and to a task being unsolvable.
> - For example, SWE-Bench (https://arxiv.org/abs/2310.06770) followed a similar approach by releasing a human-validated subset called SWE-Bench Verified (https://openai.com/index/introducing-swe-bench-verified/) so that perfect performance is attainable, which provides a stronger signal to developers when designing agents. Just as SWE-Bench-Verified doesn’t contain “negative examples” that are unsolvable, neither does CORE-Bench.
>
> > Please provide code which can be run locally, for free, and in a reasonable amount of time which would allow a reader to test a part of your benchmark.
>
> Please see our above response.
>
> > Please provide additional justification for the decision not to include any "negative examples," i.e., papers which are not reproducible, in the benchmark.
>
> Please see our above response.
>
> > The claim of the header of Section 4.1 is not exactly correct, as the AutoGPT baseline actually performs slightly better on the medium vs. the easy level. Can the authors provide some explanation for this?
>
> Thank you. We updated the heading to make it precise. Although CORE-Bench-Medium is strictly harder than CORE-Bench-Easy, since it requires the extra step of running a Docker command to reproduce the code outputs, in practice, agents with strong underlying models like GPT-4o perform very similarly on the two tasks. Our updated heading reflects this.

---

> > ### Comment · Reviewer_t3v5 · 2024-11-14
> >
> > Thanks to the authors for their response. I believe the decision to not include non-reproducible benchmarks is adequately justified, and the cost for running a proof-of-concept is also at an acceptable level. I have changed my review to recommend acceptance.

---

### Review · Reviewer_SjHV · 2024-10-15

**Summary Of Contributions:**

This paper defines a new benchmark, CORE-Bench, that measures the ability of LLM agents to effectively imitate the reproduction of results from studies known to be reproducible from three domains, all hosted on CodeOcean.com. The paper additionally proposes two baseline measures, the first using a (mostly) vanilla instantiation of AutoGPT, the second an agent (called CORE-Agent) customized specifically for this task. The authors perform detailed qualitative error analysis on where current models fail.

**Audience:**

Yes

**Broader Impact Concerns:**

None.

**Claims And Evidence:**

No

**Requested Changes:**

1. Scoping down the claims of the paper to be more in line with what the paper actually demonstrates
2. Some degree of quantification of the error analysis - how common were the common failure points?

**Strengths And Weaknesses:**

## Strengths:

S1: This is a timely task, and I really appreciate the task definition: being able to replicate results is indeed important, and I think this paper thoughtfully defines a part of that that should theoretically be solvable by agents, and designs a benchmark around it.

S2: Continuing on task setup, the setup of the overlapping layers of difficulty is really solid. That mistakes in Easy are a subset of the ones that can be made in medium, and so on, makes this setup a really good basis for doing diagnostic work on where models fail, and how to make them better.

S3: I really appreciate the authors' willingness to look at particular instances of error cases that these models make. The kind of qualitative analysis in this paper is really welcome. That said, I think it would be great to see a systematic breakdown of what kinds of tasks and stages models fail on. This is clearly something the authors have already done, at least in part - they mention that models sometimes fail in installing dependencies, or fail to locate the correct file from which to evaluate results. I'd love to see the percentages of which models break down at which steps. This would be a perfect opportunity to use e.g. a Sankey diagram per model to display what the common failure modes of models are, so that users of the benchmark can easily diagnose what high-impact fixes for their agents might be. I'd also really appreciate some degree of breakdown of what the specific questions are and evaluation on those - how did models fare on identifying axes vs specific points? The authors currently break this down by modality, but no further.

S4: The counterfactual evaluation for things like compute budget showed really thoughtful experimentation and verifying that the choices made here were reasonable.

## Weaknesses

 I have a couple here, but I think only the first one is a real issue.

W1: This paper's intended scope, as defined in the research question on page 3, is much broader than the paper eventually answers. This paper represents a solid step towards that research question, but to be clear, it evaluates models on their ability to reproduce results that are _already known to be reproducible._ To answer the question as posed, there are a number of questions that this paper doesn't answer: what is the models' behavior in cases where the research is not in fact reproducible when attempting to run the provided code? How would such a system differentiate between agent failures and more fundamental reproducibility failures?

These are questions that could reasonably be argued to be beyond the scope of this paper, but in claiming that this work has "real world utility," in that a model that is strong on this benchmark would help authors "verify their work's reproducibility," the authors assert that those questions are functionally answered. I think scoping down the claim of this paper, and making clear that this is a necessary, but not sufficient step towards automated verification of reproducibility, would render this paper no less strong in its provided benchmark and baselines.

W2: I don't understand the point about how this code being from public repositories mitigates concerns about contamination. Even if they can be periodically updated, what is the number of repositories that are, especially after paper submission? Could the models that underlie the agents for this benchmark not also be retrained?

W3: The difference between CORE-Bench Easy and Medium is very small (and reflected in a correspondingly small accuracy drop), and the difference between Medium and Hard is really large. The paper notes this, but I think the utility of this benchmark and results obtained on it would be helped by understanding _where_ models fail as part of the benchmark - if it's in dependency management, the benchmark could verify the environment before experiments are run; similar steps could be taken for other large steps in the process. While it's useful to know that models fail given the whole repository and instructions, knowing why and when they fail would make this benchmark much more useful as a diagnostic for model performance.

---

> ### Author Response · Authors · 2024-10-30
> **Thank you for your review**
>
> We thank the reviewer for their thoughtful review. We have incorporated several points of feedback into our updated manuscript.
>
> > S3: I really appreciate the authors' willingness to look at particular instances of error cases that these models make. The kind of qualitative analysis in this paper is really welcome. That said, I think it would be great to see a systematic breakdown of what kinds of tasks and stages models fail on. This is clearly something the authors have already done, at least in part - they mention that models sometimes fail in installing dependencies, or fail to locate the correct file from which to evaluate results. I'd love to see the percentages of which models break down at which steps. This would be a perfect opportunity to use e.g. a Sankey diagram per model to display what the common failure modes of models are, so that users of the benchmark can easily diagnose what high-impact fixes for their agents might be. I'd also really appreciate some degree of breakdown of what the specific questions are and evaluation on those - how did models fare on identifying axes vs specific points? The authors currently break this down by modality, but no further.
>
> We appreciate the suggestion for a more detailed analysis of failure cases on CORE-Bench-Hard, which we provide later on in this response. Since there are tens of different templates for task questions, it is difficult to construct a breakdown of performance on different questions, but we added more examples of task questions to the appendix to give readers a better understanding of the benchmark construction.
>
> > W1: This paper's intended scope, as defined in the research question on page 3, is much broader than the paper eventually answers. This paper represents a solid step towards that research question, but to be clear, it evaluates models on their ability to reproduce results that are already known to be reproducible. To answer the question as posed, there are a number of questions that this paper doesn't answer: what is the models' behavior in cases where the research is not in fact reproducible when attempting to run the provided code? How would such a system differentiate between agent failures and more fundamental reproducibility failures?
>
> We acknowledge the lack of clarity of our scope in our current draft, and appreciate the feedback that the scope and real world utility should be made more clear. In response, we have updated the introduction, section 2.1, and the conclusion. A brief overview of what we clarified is included below.
> - We decompose the task of verifying computational reproducibility into two sub-tasks: code reproducibility and result reproducibility. This paper and benchmark focuses on code reproducibility, or running the code and obtaining the results the capsule is supposed to produce, whether or not those match the results reported in the paper (in fact, 30/90 included papers are not result-reproducible). We focus on code reproducibility since it is by far the more time consuming part for a human.
> - Specifically, the benchmark is used to evaluate if agents can automate code reproducibility, as opposed to verifying if papers are reproducible. If in the future a high performing agent is created, it could also help with verifying that existing papers are reproducible, but we are far off from that point.
> - Agents developed for this benchmark could still be useful to automate reproductions of other papers —once a paper's results are reproduced, verifying that it matches the results in a paper (results reproducibility) is much easier. This claim is not meant to take away from the other important questions the reviewer raises, and we agree important future work remains, which we added to the conclusion.
> - We did not include code-irreproducible capsules because agents would just fail since the task questions would be impossible to answer. Indeed, we want all benchmark tasks to be human-verified as solvable that way any failure of reproducing a paper can be attributed to a failure of the agent and to a task being unsolvable.
> - For example, SWE-Bench (https://arxiv.org/abs/2310.06770) followed a similar approach by releasing a human-validated subset called SWE-Bench Verified (https://openai.com/index/introducing-swe-bench-verified/) so that perfect performance is attainable, which provides a stronger signal to developers when designing agents. Just as SWE-Bench-Verified doesn’t contain “negative examples” that are unsolvable, neither does CORE-Bench.

---

> > ### Author Response · Authors · 2024-10-30
> >
> > > W2: I don't understand the point about how this code being from public repositories mitigates concerns about contamination. Even if they can be periodically updated, what is the number of repositories that are, especially after paper submission? Could the models that underlie the agents for this benchmark not also be retrained?
> >
> > Thank you for pointing this out – we agree that contamination remains a limitation and that the code being from public repositories does not directly address this. Our point was that using public repositories that are updated over time with fresh tasks makes it easier to update the benchmark in the future, but we acknowledge that this only means our methods have the potential to address contamination via future work (such as by using model versions with a certain training cutoff date and only evaluating them on papers released after that date).
> >
> > Still, we have taken some steps to address contamination, such as requiring developers to decrypt our test set questions so they aren’t easily scraped, developing a private held-out test set that we will use to evaluate top performing agents on CORE-Bench, and using fixed model versions to prevent any retraining from affecting evaluations. However, we updated the wording to acknowledge that contamination remains a concern.
> >
> > > W3: The difference between CORE-Bench Easy and Medium is very small (and reflected in a correspondingly small accuracy drop), and the difference between Medium and Hard is really large. The paper notes this, but I think the utility of this benchmark and results obtained on it would be helped by understanding where models fail as part of the benchmark - if it's in dependency management, the benchmark could verify the environment before experiments are run; similar steps could be taken for other large steps in the process. While it's useful to know that models fail given the whole repository and instructions, knowing why and when they fail would make this benchmark much more useful as a diagnostic for model performance.
> >
> > Based on the reviewer’s suggestion, we ran a qualitative analysis of the failure cases on all levels of the benchmark for the best-performing agent, and included the results in the updated manuscript. A brief summary:
> > - All of the failures by CORE-Agent with GPT-4o on CORE-Bench-Easy were caused by an inability to retrieve the relevant information from the code outputs (40.00% of tasks), since the task setup already provides agents with the code outputs.
> > - Although CORE-Bench-Medium is strictly harder than CORE-Bench-Easy, since it requires the extra step of running a Docker command to reproduce the code outputs, in practice, CORE-Agent with GPT-4o did almost as well, and failures (42.22% of tasks) were also due to incorrectly retrieving the relevant information from the code outputs.
> > - On CORE-Bench-Hard,
> >     - The agent successfully completed 22.22% of all tasks by answering all of the task questions correctly.
> >     - On 17.78% of tasks, the agent was able to properly install all dependencies and run the code but was unable to retrieve the relevant information from the results to answer the task questions.
> >     - On 15.56% of tasks, the agent successfully installed the dependencies but was unable to run the code properly, usually because it did not run the correct command to reproduce the code or it did not run the command from the proper directory.
> >     - On the remaining 44.44% of tasks, the agent was unable to properly install the dependencies usually due to installing the incorrect version of libraries and hitting the cost limit. We included this analysis in the updated manuscript.
> >
> > > Scoping down the claims of the paper to be more in line with what the paper actually demonstrates
> >
> > We clarified Section 2.1 of the paper to address this concern.
> >
> > > Some degree of quantification of the error analysis - how common were the common failure points?
> >
> > We performed a qualitative analysis.

---

### Decision · Action_Editor_GBic · 2024-11-23

**Recommendation:** Accept with minor revision

**Comment:**

SjHV accurately points out
> "What this measures is not whether AI agents can automate computational reproducibility, but whether computational reproducibility can be verified by AI agents, which is both a weaker claim, and results in less useful AI agents"

while
> "primarily problems of framing"

The authors failed to appreciate this nuance in their rebuttal.  The work otherwise appears to be technically correct. It is therefore my responsibility to ensure that the revision correctly rescopes to match the results.

**Audience:**

Reproducibility and LLM Agents

**Claims And Evidence:**

The aim of this work is to demonstrate that LLM agents can be used to validate the reproducibility of research.  The basic approach is the introduction of a dataset and two baselines (AutoGPT and their own CORE-Agent).  The work assumes the presence of data and code (R or Python) which the agent can then execute for answering questions about the work.